# TRS: Transferability Reduced Ensemble via Promoting Gradient Diversity and Model Smoothness

**Zhuolin Yang**[1*]     **Linyi Li**[1*]     **Xiaojun Xu**[1*]     **Shiliang Zuo**[1]
**Qian Chen**[2]     **Benjamin Rubinstein**[3]     **Pan Zhou**[4]     **Ce Zhang**[5]     **Bo Li**[1]

[1] University of Illinois Urbana-Champaign [2] Tencent Inc. [3] University of Melbourne
[4] Huazhong University of Science and Technology [5] ETH Zurich
{zhuolin5, linyi2, xiaojun3, szuo3, lbo}@illinois.edu
qianchen@tencent.com
benjamin.rubinstein@unimelb.edu.au
panzhou@hust.edu.cn
ce.zhang@inf.ethz.ch

## Abstract

*Adversarial Transferability* is an intriguing property – adversarial perturbation crafted against one model is also effective against another model, while these models are from different model families or training processes. To better protect ML systems against adversarial attacks, several questions are raised: *what are the sufficient conditions for adversarial transferability and how to bound it? Is there a way to reduce the adversarial transferability in order to improve the robustness of an ensemble ML model?* To answer these questions, in this work we first theoretically analyze and outline sufficient conditions for adversarial transferability between models; then propose a practical algorithm to reduce the transferability between base models within an ensemble to improve its robustness. Our theoretical analysis shows that only promoting the orthogonality between gradients of base models is not enough to ensure low transferability; in the meantime, the model smoothness is an important factor to control the transferability. We also provide the lower and upper bounds of adversarial transferability under certain conditions. Inspired by our theoretical analysis, we propose an effective **T**ransferability **R**educed **S**mooth (TRS) ensemble training strategy to train a robust ensemble with low transferability by enforcing both gradient orthogonality and model smoothness between base models. We conduct extensive experiments on TRS and compare with 6 state-of-the-art ensemble baselines against 8 whitebox attacks on different datasets, demonstrating that the proposed TRS outperforms all baselines significantly.

## 1   Introduction

Machine learning systems, especially those based on deep neural networks (DNNs), have been widely applied in numerous applications [27, 18, 46, 10]. However, recent studies show that DNNs are vulnerable to adversarial examples, which are able to mislead DNNs by adding small magnitude of perturbations to the original instances [47, 17, 54, 52]. Several attack strategies have been proposed so far to generate such adversarial examples in both digital and physical environments [36, 32, 51, 53, 15, 28]. Intriguingly, though most attacks require access to the target models (whitebox attacks), several studies show that adversarial examples generated against one model are able to *transferably*

---

*The authors contributed equally.

35th Conference on Neural Information Processing Systems (NeurIPS 2021).

attack another target model with high probability, giving rise to blackbox attacks [39, 41, 31, 30, 57]. This property of *adversarial transferability* poses great threat to DNNs.

Some work have been conducted to understand *adversarial transferability* [48, 33, 12]. However, a rigorous theoretical analysis or explanation for transferability is still lacking in the literature. In addition, although developing robust ensemble models to limit transferability shows great potential towards practical robust learning systems, only *empirical* observations have been made in this line of research [38, 23, 56]. *Can we deepen our theoretical understanding on transferability? Can we take advantage of rigorous theoretical understanding to reduce the adversarial transferability and therefore generate robust ensemble ML models?*

In this paper, we focus on these two questions. From the theoretical side, we are interested in the sufficient conditions under which the adversarial transferability can be *lower bounded* and *upper bounded*. Our theoretical arguments provides the *first* theoretical interpretation for the sufficient conditions of transferability. Intuitively, as illustrated in Figure 1, we show that the commonly used gradient orthogonality (low cosine similarity) between learning models [12] cannot directly imply low adversarial transferability; on the other hand, orthogonal and smoothed models would limit the transferability. In particular, we prove that the *gradient similarity* and *model smoothness* are the key factors that both contribute to the adversarial transferability, and smooth models with orthogonal gradients can guarantee low transferability.

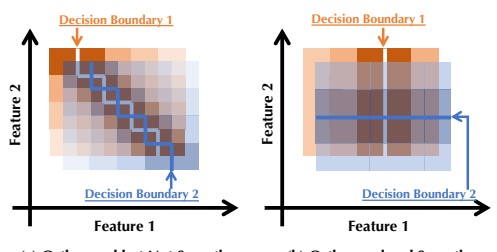

Figure 1: An illustration of the relationship between *adversarial transferability*, *gradient orthogonality*, and *model smoothness*. (a) Gradient orthogonality alone cannot minimize transferability as the decision boundaries between two classifiers can be arbitrarily close yet orthogonal almost everywhere; (b) Gradient orthogonality with model smoothness provides a stronger guarantee on model diversity, as our theorems will show.

Under an empirical lens, inspired by our theoretical analysis, we propose a simple yet effective approach, **T**ransferability **R**educed **S**mooth (TRS) ensemble to limit adversarial transferability between base models within an ensemble and therefore improve its robustness. In particular, we reduce the loss gradient similarity between models as well as enforce the smoothness of models to introduce global model orthogonality.

We conduct extensive experiments to evaluate TRS in terms of the model robustness against different strong white-box and blackbox attacks following the robustness evaluation procedures [5, 6, 49], as well as its ability to limit transferability across the base models. We compare the proposed TRS with existing state-of-the-art baseline ensemble approaches such as ADP [38], GAL [23], and DVERGE [56] on MNIST, CIFAR-10, and CIFAR-100 datasets, and we show that (1) TRS achieves the state-of-the-art ensemble robustness, outperforming others by a large margin; (2) TRS achieves efficient training; (3) TRS effectively reduces the transferability among base models within an ensemble which indicates its robustness against whitebox and blackbox attacks; (4) Both loss terms in TRS contribute to the ensemble robustness by constraining different sufficient conditions of adversarial transferability.

**Contributions.** In this paper, we make the first attempt towards theoretical understanding of adversarial transferability, and provide practical approach for developing robust ML ensembles.

(*1*) We provide a general theoretical analysis framework for adversarial transferability. We prove the lower and upper bounds of adversarial transferability. Both bounds show that the gradient similarity and model smoothness are the key factors contributing to the adversarial transferability, and smooth models with orthogonal gradients can guarantee low transferability.

(*2*) We propose a simple yet effective approach TRS to train a robust ensemble by jointly reducing the loss gradient similarity between base models and enforcing the model smoothness. The code is publicly available[2].

(*3*) We conduct extensive experiments to evaluate TRS in terms of model robustness under different attack settings, showing that TRS achieves the state-of-the-art ensemble robustness and outperforms other baselines by a large margin. We also conduct ablation studies to further understand the contribution of different loss terms and verify our theoretical findings.

---

[2]https://github.com/AI-secure/Transferability-Reduced-Smooth-Ensemble

**Related Work**

The adversarial transferability between different ML models is an intriguing research direction. Papernot et al. [40] explored the limitation of adversarial examples and showed that, while some instances are more difficult to manipulate than the others, these adversarial examples usually transfer from one model to another. Demontis et al. [12] later analyzed transferability for both evasion and poisoning attacks. Tramèr et al. [48] empirically investigated the subspace of adversarial examples that enables transferability between different models: though their results provide a non-zero probability guarantee on the transferability, they did not quantify the probability of adversarial transferability.

Leveraging the transferability, different blackbox attacks have been proposed [41, 28, 15, 9]. To defend against these transferability based attacks, Pang et al. [38] proposed a class entropy based adaptive diversity promoting approach to enhance the ML ensemble robustness. Recently, Yang et al. [56] proposed DVERGE, a robust ensemble training approach that diversifies the non-robust features of base models via an adversarial training objective function. However, these approaches do not provide theoretical justification for adversarial transferability, and there is still room to improve the ML ensemble robustness based on in-depth understanding on the sufficient conditions of transferability. In this paper, we aim to provide a theoretical understanding of transferability, and empirically compare the proposed robust ML ensemble inspired by our theoretical analysis with existing approaches to push for a tighter empirical upper bound for the ensemble robustness.

## 2 Transferability of Adversarial Perturbation

In this section, we first introduce preliminaries, and then provide the upper and lower bounds of adversarial transferability by connecting adversarial transferability with different characteristics of models theoretically, which, in the next section, will allow us to explicitly minimize transferability by enforcing (or rewarding) certain properties of models.

**Notations.** We consider neural networks for classification tasks. Assume there are $C$ classes, and let $\mathcal{X}$ be the *input space* of the model with $\mathcal{Y} = \{1, 2, \ldots, C\}$ the set of prediction classes (i.e., labels). We model the neural network by a mapping function $\mathcal{F} : \mathcal{X} \to \mathcal{Y}$. We will study the transferability between two models $\mathcal{F}$ and $\mathcal{G}$. For brevity, hereinafter we mainly show the derived notations for $\mathcal{F}$ and notations for $\mathcal{G}$ are similar. Let the *benign* data $(x, y)$ follow an unknown distribution $\mathcal{D}$ supported on $(\mathcal{X}, \mathcal{Y})$, and $\mathcal{P}_{\mathcal{X}}$ denote the marginal distribution on $\mathcal{X}$.

For a given input $x \in \mathcal{X}$, the classification model $\mathcal{F}$ first predicts the confidence score for each label $y \in \mathcal{Y}$, denoted as $f_y(x)$. These confidence scores sum up to 1, i.e., $\sum_{y \in \mathcal{Y}} f_y(x) = 1, \forall x \in \mathcal{X}$. The model $\mathcal{F}$ will predicts the label with highest confidence score: $\mathcal{F}(x) = \arg\max_{y \in \mathcal{Y}} f_y(x)$.

For model $\mathcal{F}$, there is usually a model-dependent loss function $\ell_{\mathcal{F}} : \mathcal{X} \times \mathcal{Y} \to \mathbb{R}_+$, which is the composition of a differentiable training loss (e.g., cross-entropy loss) $\ell$ and the model's confidence score $f(\cdot)$: $\ell_{\mathcal{F}}(x, y) := \ell(f(x), y), (x, y) \in (\mathcal{X}, \mathcal{Y})$. We further assume that $\mathcal{F}(x) = \arg\min_{y \in \mathcal{Y}} \ell_{\mathcal{F}}(x, y)$, i.e., the model predicts the label with minimum loss. This holds for common training losses.

In this paper, by default we will focus on models that are well-trained on the benign dataset, and such models are the most commonly encountered in practice, so their robustness is paramount. This means we will focus on the *low risk* classifiers, which we will formally define in Section 2.1.

*How should we define an adversarial attack?* For the threat model, we consider the attacker that adds an $\ell_p$ norm bounded perturbation to data instance $x \in \mathcal{X}$. In practice, there are two types of attacks, *untargeted attacks* and *targeted attacks*. The definition of adversarial transferability is slightly different under these attacks [33], and we consider both in our analysis.

**Definition 1** (Adversarial Attack). *Given an input $x \in \mathcal{X}$ with true label $y \in \mathcal{Y}$, $\mathcal{F}(x) = y$. (1) An untargeted attack crafts $\mathcal{A}_U(x) = x + \delta$ to maximize $\ell_{\mathcal{F}}(x + \delta, y)$ where $\|\delta\|_p \leq \epsilon$. (2) A targeted attack with target label $y_t \in \mathcal{Y}$ crafts $\mathcal{A}_T(x) = x + \delta$ to minimize $\ell_{\mathcal{F}}(x + \delta, y_t)$ where $\|\delta\|_p \leq \epsilon$.*

In this definition, $\epsilon$ is a pre-defined *attack radius* that limits the power of the attacker. We may refer to $\{\delta : \|\delta\|_p \leq \epsilon\}$ as the perturbation ball. The goal of the untargeted attack is to maximize the loss of the target model against its true label $y$. The goal of the targeted attack is to minimize the loss towards its adversarial target label $y_t$.

*How do we formally define that an attack is effective?*

**Definition 2** (($\alpha, \mathcal{F}$)-Effective Attack). *Consider a input $x \in \mathcal{X}$ with true label $y \in \mathcal{Y}$. An attack is $(\alpha, \mathcal{F})$-effective in untargeted scenario if $\Pr\left(\mathcal{F}(\mathcal{A}_U(x)) \neq y\right) \geq 1 - \alpha$. An attack is $(\alpha, \mathcal{F})$-effective in targeted scenario (with class target $y_t$) if $\Pr\left(\mathcal{F}(\mathcal{A}_T(x)) = y_t\right) \geq 1 - \alpha$.*

This definition captures the requirement that an adversarial instance generated by an effective attack strategy is able to mislead the target classification model (e.g. $\mathcal{F}$) with certain probability $(1 - \alpha)$. The smaller the $\alpha$ is, the more effective the attack is. In practice, this implies that on a finite sample of targets, the attack success is frequent but not absolute. Note that the definition is general for both whitebox [1, 12, 5] and blackbox attacks [42, 4].

## 2.1 Model Characteristics

*Given two models $\mathcal{F}$ and $\mathcal{G}$, what are the characteristics of $\mathcal{F}$ and $\mathcal{G}$ that have impact on transferability under a given attack strategy?* Intuitively, the more similar these two classifers are, the larger the transferability would be. However, *how can we define "similar" and how can we rigorously connect it to transferability?* To answer these questions, we will first define the risk and empirical risk for a given model to measure its performance on benign test data. Then, as the DNNs are differentiable, we will define model similarity based on their gradients. We will then derive the lower and upper bounds of adversarial transferability based on the defined model risk and similarity measures.

**Definition 3** (Risk and Empirical Risk). *For a given model $\mathcal{F}$, we let $\ell_{\mathcal{F}}$ be its model-dependent loss function. Its **risk** is defined as $\eta_{\mathcal{F}} = \Pr\left(\mathcal{F}(x) \neq y\right)$; and its **empirical risk** is defined as $\xi_{\mathcal{F}} = \mathbb{E}\left[\ell_{\mathcal{F}}(x, y)\right]$.*

The *risk* represents the model's error rate on benign test data, while the *empirical risk* is a non-negative value that also indicates the inaccuracy. For both of them, higher value means worse performance on the benign test data. The difference is that, the risk has more intuitive meaning, while the empirical risk is differentiable and is actually used during model training.

**Definition 4** (Loss Gradient Similarity). *The lower loss gradient similarity $\underline{\mathcal{S}}$ and upper loss gradient similarity $\overline{\mathcal{S}}$ between two differentiable loss functions $\ell_{\mathcal{F}}$ and $\ell_{\mathcal{G}}$ is defined as:*

$$\underline{\mathcal{S}}(\ell_{\mathcal{F}}, \ell_{\mathcal{G}}) = \inf_{x \in \mathcal{X}, y \in \mathcal{Y}} \frac{\nabla_x \ell_{\mathcal{F}}(x, y) \cdot \nabla_x \ell_{\mathcal{G}}(x, y)}{\|\nabla_x \ell_{\mathcal{F}}(x, y)\|_2 \cdot \|\nabla_x \ell_{\mathcal{G}}(x, y)\|_2}, \overline{\mathcal{S}}(\ell_{\mathcal{F}}, \ell_{\mathcal{G}}) = \sup_{x \in \mathcal{X}, y \in \mathcal{Y}} \frac{\nabla_x \ell_{\mathcal{F}}(x, y) \cdot \nabla_x \ell_{\mathcal{G}}(x, y)}{\|\nabla_x \ell_{\mathcal{F}}(x, y)\|_2 \cdot \|\nabla_x \ell_{\mathcal{G}}(x, y)\|_2}.$$

The $\underline{\mathcal{S}}(\ell_{\mathcal{F}}, \ell_{\mathcal{G}})$ ($\overline{\mathcal{S}}(\ell_{\mathcal{F}}, \ell_{\mathcal{G}})$) is the minimum (maximum) cosine similarity between the gradients of the two loss functions for an input $x$ drawn from $\mathcal{X}$ with any label $y \in \mathcal{Y}$. Besides the loss gradient similarity, in our analysis we will also show that the *model smoothness* is another key characteristic of ML models that affects the model transferability.

**Definition 5.** *We call a model $\mathcal{F}$ $\beta$-smooth if* $\displaystyle\sup_{x_1, x_2 \in \mathcal{X}, y \in \mathcal{Y}} \frac{\|\nabla_x \ell_{\mathcal{F}}(x_1, y) - \nabla_x \ell_{\mathcal{F}}(x_2, y)\|_2}{\|x_1 - x_2\|_2} \leq \beta.$

This smoothness definition is commonly used in deep learning theory and optimization literature [3, 2], and is also named curvature bounds in certified robustness literature [44]. It could be interpreted as the Lipschitz bound for the model's loss function gradient. We remark that *larger $\beta$* indicates that the model is less smoother, while *smaller $\beta$* means the model is smoother. Particularly, when $\beta = 0$, the model is linear in the input space $\mathcal{X}$.

## 2.2 Definition of Adversarial Transferability

Based on the model characteristics we explored above, next we will ask: *Given two models, what is the natural and precise definition of adversarial transferability?*

**Definition 6** (Transferability). *Consider an adversarial instance $\mathcal{A}_U(x)$ or $\mathcal{A}_T(x)$ constructed against a surrogate model $\mathcal{F}$. With a given benign input $x \in \mathcal{X}$, The transferability $T_r$ between $\mathcal{F}$ and a target model $\mathcal{G}$ is defined as follows (adversarial target $y_t \in \mathcal{Y}$):*

- *Untargeted: $T_r(\mathcal{F}, \mathcal{G}, x) = \mathbb{I}[\mathcal{F}(x) = \mathcal{G}(x) = y \;\wedge\; \mathcal{F}(\mathcal{A}_U(x)) \neq y \;\wedge\; \mathcal{G}(\mathcal{A}_U(x)) \neq y].$*

- *Targeted: $T_r(\mathcal{F}, \mathcal{G}, x, y_t) = \mathbb{I}[\mathcal{F}(x) = \mathcal{G}(x) = y \;\wedge\; \mathcal{F}(\mathcal{A}_T(x)) = \mathcal{G}(\mathcal{A}_T(x)) = y_t].$*

Here we define the transferability at instance level, showing several conditions are required to satisfy for a transferable instance. For the untargeted attack, it requires that: (1) both the surrogate model

and target model make correct prediction on the benign input; and (2) both of them make incorrect predictions on the adversarial input $\mathcal{A}_U(x)$. The $\mathcal{A}_U(x)$ is generated via the untargeted attack against the surrogate model $\mathcal{F}$. For the targeted attack, it requires that: (1) both the surrogate and target model make correct prediction on benign input; and (2) both output the adversarial target $y_t \in \mathcal{Y}$ on the adversarial input $\mathcal{A}_T(x)$. The $\mathcal{A}_T(x)$ is crafted against the surrogate model $\mathcal{F}$. The predicates themselves do not require $\mathcal{A}_U$ and $\mathcal{A}_T$ to be explicitly constructed against the surrogate model $\mathcal{F}$. It will be implied by attack effectiveness (Definition 2) on $\mathcal{F}$ in theorem statements. Note that the definition here is a predicate for a specific input $x$, and in the following analysis we will mainly use its distributional version: $\Pr\left(T_r(\mathcal{F}, \mathcal{G}, x) = 1\right)$ and $\Pr\left(T_r(\mathcal{F}, \mathcal{G}, x, y_t) = 1\right)$.

## 2.3 Lower Bound of Adversarial Transferability

Based on the general definition of transferability, in this section we will analyze how to lower bound the transferability for targeted attack. The analysis for untargeted attack has a similar form and is deferred to Theorem 3 in Appendix A.

**Theorem 1** (Lower Bound on Targeted Attack Transferability). *Assume both models $\mathcal{F}$ and $\mathcal{G}$ are $\beta$-smooth. Let $\mathcal{A}_T$ be an $(\alpha, \mathcal{F})$-effective targeted attack with perturbation ball $\|\delta\|_2 \leq \epsilon$ and target label $y_t \in \mathcal{Y}$. The transferabiity can be lower bounded by*

$$\Pr\left(T_r(\mathcal{F}, \mathcal{G}, x, y_t) = 1\right) \geq (1-\alpha) - (\eta_\mathcal{F} + \eta_\mathcal{G}) - \frac{\epsilon(1+\alpha) + c_\mathcal{F}(1-\alpha)}{c_\mathcal{G} + \epsilon} - \frac{\epsilon(1-\alpha)}{c_\mathcal{G} + \epsilon}\sqrt{2 - 2\underline{\mathcal{S}}(\ell_\mathcal{F}, \ell_\mathcal{G})},$$

*where*

$$c_\mathcal{F} = \max_{x \in \mathcal{X}} \frac{\min_{y \in \mathcal{Y}} \ell_\mathcal{F}(\mathcal{A}_T(x), y) - \ell_\mathcal{F}(x, y_t) + \beta\epsilon^2/2}{\|\nabla_x \ell_\mathcal{F}(x, y_t)\|_2}, c_\mathcal{G} = \min_{x \in \mathcal{X}} \frac{\min_{y \in \mathcal{Y}} \ell_\mathcal{G}(\mathcal{A}_T(x), y) - \ell_\mathcal{G}(x, y_t) - \beta\epsilon^2/2}{\|\nabla_x \ell_\mathcal{G}(x, y_t)\|_2}.$$

*Here $\eta_\mathcal{F}, \eta_\mathcal{G}$ are the* risks *of models $\mathcal{F}$ and $\mathcal{G}$ respectively.*

We defer the complete proof in Appendix C. In the proof, we first use a Taylor expansion to introduce the gradient terms, then relate the dot product with cosine similarity of the loss gradients, and finally use Markov's inequality to derive the misclassification probability of $\mathcal{G}$ to complete the proof.

**Implications.** In Theorem 1, the only term which correlates both $\mathcal{F}$ and $\mathcal{G}$ is $\underline{\mathcal{S}}(\ell_\mathcal{F}, \ell_\mathcal{G})$, while all other terms depend on individual models $\mathcal{F}$ or $\mathcal{G}$. Thus, we study the relation between $\underline{\mathcal{S}}(\ell_\mathcal{F}, \ell_\mathcal{G})$ and $\Pr\left(T_r(\mathcal{F}, \mathcal{G}, x, y_t) = 1\right)$.

Note that since $\beta$ is small compared with the perturbation radius $\epsilon$ and the gradient magnitude $\|\nabla_x \ell_\mathcal{G}\|_2$ in the denominator is relatively large, the quantity $c_\mathcal{G}$ is small. Moreover, $1 - \alpha$ is large since the attack is typically effective against $\mathcal{F}$. Thus, $\Pr\left(T_r(\mathcal{F}, \mathcal{G}, x, y_t) = 1\right)$ has the form $C - k\sqrt{1 - \underline{\mathcal{S}}(\ell_\mathcal{F}, \ell_\mathcal{G})}$, where $C$ and $k$ are both positive constants. We can easily observe the *positive* correlation between the loss gradients similarity $\underline{\mathcal{S}}(\ell_\mathcal{F}, \ell_\mathcal{G})$, and lower bound of adversarial transferability $\Pr\left(T_r(\mathcal{F}, \mathcal{G}, x, y_t) = 1\right)$.

In the meantime, note that when $\beta$ increases (i.e., model becomes less smooth), in the transferability lower bound $C - k\sqrt{1 - \underline{\mathcal{S}}(\ell_\mathcal{F}, \ell_\mathcal{G})}$, the $C$ decreases and $k$ increase. As a result, the lower bounds in Theorem 1 decreases, which implies that when model becomes less smoother (i.e., $\beta$ becomes larger), the transferability lower bounds become looser for both targeted and untargeted attacks. In other words, *when the model becomes smoother, the correlation between loss gradients similarity and lower bound of transferability becomes stronger*, which motivates us to constrain the model smoothness to increase the effect of limiting loss gradients similarity.

In addition to the $\ell_p$-bounded attacks, we also derive a transferability lower bound for general attacks whose magnitude is bounded by total variance distance of data distributions. We defer the detail analysis and discussion to Appendix B.

## 2.4 Upper Bound of Adversarial Transferability

We next aim to upper bound the adversarial transferability. The upper bound for target attack is shown below; and the one for untargeted attack has a similar form in Theorem 4 in Appendix A.

**Theorem 2** (Upper Bound on Targeted Attack Transferability). *Assume both models $\mathcal{F}$ and $\mathcal{G}$ are $\beta$-smooth with gradient magnitude bounded by $B$, i.e., $\|\nabla_x \ell_\mathcal{F}(x, y)\| \leq B$ and $\|\nabla_x \ell_\mathcal{G}(x, y)\| \leq B$ for any $x \in \mathcal{X}, y \in \mathcal{Y}$. Let $\mathcal{A}_T$ be an $(\alpha, \mathcal{F})$-effective targeted attack with perturbation ball $\|\delta\|_2 \leq \epsilon$*

*and target label $y_t \in \mathcal{Y}$. When the attack radius $\epsilon$ is small such that $\ell_{\min} - \epsilon B \left( 1 + \sqrt{\frac{1+\overline{\mathcal{S}}(\ell_\mathcal{F}, \ell_\mathcal{G})}{2}} \right) -$*

*$\beta \epsilon^2 > 0$, the transferability can be upper bounded by*

$$\Pr\left(T_r(\mathcal{F}, \mathcal{G}, x, y_t) = 1\right) \leq \frac{\xi_\mathcal{F} + \xi_\mathcal{G}}{\ell_{\min} - \epsilon B \left( 1 + \sqrt{\frac{1+\overline{\mathcal{S}}(\ell_\mathcal{F}, \ell_\mathcal{G})}{2}} \right) - \beta \epsilon^2},$$

*where $\ell_{\min} = \min\limits_{x \in \mathcal{X}} \left(\ell_\mathcal{F}(x, y_t), \ell_\mathcal{G}(x, y_t)\right)$. Here $\xi_\mathcal{F}$ and $\xi_\mathcal{G}$ are the* empirical risks *of models $\mathcal{F}$ and $\mathcal{G}$ respectively, defined relative to a differentiable loss.*

We defer the complete proof to Appendix D. In the proof, we first take a Taylor expansion on the loss function at $(x, y)$, then use the fact that the attack direction will be dissimilar with at least one of the model gradients to upper bound the transferability probability.

**Implications.** In Theorem 2, we observe that along with the increase of $\overline{\mathcal{S}}(\ell_\mathcal{F}, \ell_\mathcal{G})$, the denominator decreases and henceforth the upper bound increases. Therefore, $\overline{\mathcal{S}}(\ell_\mathcal{F}, \ell_\mathcal{G})$—upper loss gradient similarity and the upper bound of transferability probability is positively correlated. This tendency is the same as that in the lower bound. Note that $\alpha$ does not appear in upper bounds since only completely successful attacks ($\alpha = 0\%$) needs to be considered here to upper bound the transferability.

Meanwhile, when the model becomes smoother (i.e., $\beta$ decreases), the transferability upper bound decreases and becomes tighter. This implication again motivates us to constrain the model smoothness. We further observe that smaller magnitude of gradient, i.e., $B$, also helps to tighten the upper bound. We will regularize both $B$ and $\beta$ to increase the effect of constraining loss gradients similarity.

Note that the lower bound and upper bound jointly show smaller $\beta$ leads to a reduced gap between lower and upper bounds and thus a stronger correlation between loss gradients similarity and transferabiltiy. Therefore, it is important to *both* constrain gradient similarity and increase model smoothness (decrease $\beta$) to reduce model transferability and improve ensemble robustness.

## 3 Improving Ensemble Robustness via Transferability Minimization

Motivated by our theoretical analysis, we propose a lightweight yet effective robust ensemble training approach, **T**ransferability **R**educed **S**mooth (TRS), to reduce the transferability among base models by enforcing *low loss gradient similarity* and *model smoothness* at the same time.

### 3.1 TRS Regularizer

In practice, it is challenging to directly regularize the model smoothness. Luckily, inspired from deep learning theory and optimization [14, 37, 45], succinct $\ell_2$ regularization on the gradient terms $\|\nabla_x \ell_\mathcal{F}\|_2$ and $\|\nabla_x \ell_\mathcal{G}\|_2$ can reduce the magnitude of gradients and thus improve **model smoothness**. For example, for common neural networks, the smoothness can be upper bounded via bounding the $\ell_2$ magnitude of gradients [45, Corollary 4]. An intuitive explanation is that, the $\ell_2$ regularization on the gradient terms reduces the magnitude of model's weights, thus limits its changing rate when non-linear activation functions are applied to the neural network model. However, we find that directly regularizing the loss gradient magnitude with $\ell_2$ norm is not enough, since a vanilla $\ell_2$ regularizer such as $\|\nabla_x \ell_\mathcal{F}\|_2$ will only focus on the local region at data point $x$, while it is required to ensure the model smoothness over a large decision region to control the adversarial transferability based on our theoretical analysis.

To address this challenge, we propose a min-max framework to regularize the "support" instance $\hat{x}$ with "worst" smoothness in the neighborhood region of data point $x$, which results in the following model smoothness loss:

$$\mathcal{L}_{\text{smooth}}(\mathcal{F}, \mathcal{G}, x, \delta) = \max_{\|\hat{x} - x\|_\infty \leq \delta} \|\nabla_{\hat{x}} \ell_\mathcal{F}\|_2 + \|\nabla_{\hat{x}} \ell_\mathcal{G}\|_2 \tag{1}$$

where $\delta$ refers to the radius of the $\ell_\infty$ ball around instance $x$ within which we aim to ensure the model to be smooth. In practice, we leverage projection gradient descent optimization to search for support instances $\hat{x}$ for optimization. This model smoothness loss can be viewed as promoting margin-wise smoothness, i.e., improving the margin between nonsmooth decision boundaries and data point $x$. Another option is to promote point-wise smoothness that only requires the loss landscape

at data point $x$ itself to be smooth. We compare the ensemble robustness of the proposed min-max framework which promotes the margin-wise smoothness with the naïve baseline which directly applies $\ell_2$ regularization on each model loss gradient terms to promote the point-wise smoothness (i.e. Cos-$\ell_2$) in Section 4.

Given trained "smoothed" base models, we also decrease the model **loss gradient similarity** to reduce the overall adversarial transferability between base models. Among various metrics which measure the similarity between the loss gradients of base model $\mathcal{F}$ and $\mathcal{G}$, we find that the vanilla cosine similarity metric, which is also used in [23], may lead to certain concerns. By minimizing the cosine similarity between $\nabla_x \ell_{\mathcal{F}}$ and $\nabla_x \ell_{\mathcal{G}}$, the optimal case implies $\nabla_x \ell_{\mathcal{F}} = -\nabla_x \ell_{\mathcal{G}}$, which means two models have contradictory (rather than diverse) performance on instance $x$ and thus results in turbulent model functionality. Considering this challenge, we leverage the absolute value of cosine similarity between $\nabla_x \ell_{\mathcal{F}}$ and $\nabla_x \ell_{\mathcal{G}}$ as *similarity loss* $\mathcal{L}_{\text{sim}}$ and its optimal case implies orthogonal loss gradient vectors. For simplification, we will always use the absolute value of the gradient cosine similarity as the indicator of *gradient similarity* in our later description and evaluation.

Based on our theoretical analysis and particularly the model *loss gradient similarity* and *model smoothness* optimization above, we propose TRS regularizer for model pair $(\mathcal{F}, \mathcal{G})$ on input $x$ as:

$$\mathcal{L}_{\text{TRS}}(\mathcal{F}, \mathcal{G}, x, \delta) = \lambda_a \cdot \mathcal{L}_{\text{sim}} + \lambda_b \cdot \mathcal{L}_{\text{smooth}}$$

$$= \lambda_a \cdot \left| \frac{(\nabla_x \ell_{\mathcal{F}})^\top (\nabla_x \ell_{\mathcal{G}})}{\|\nabla_x \ell_{\mathcal{F}}\|_2 \cdot \|\nabla_x \ell_{\mathcal{G}}\|_2} \right| + \lambda_b \cdot \left[ \max_{\|\hat{x} - x\|_\infty \leq \delta} \|\nabla_{\hat{x}} \ell_{\mathcal{F}}\|_2 + \|\nabla_{\hat{x}} \ell_{\mathcal{G}}\|_2 \right].$$

Here $\nabla_x \ell_{\mathcal{F}}$ and $\nabla_x \ell_{\mathcal{G}}$ refer to the loss gradient vectors of base models $\mathcal{F}$ and $\mathcal{G}$ on input $x$, and $\lambda_a, \lambda_b$ the weight balancing parameters.

In Section 4, backed up by extensive empirical evaluation, we will systematically show that the local min-max training and the absolute value of the cosine similarity between the model loss gradients significantly improve the ensemble model robustness with negligible performance drop on benign accuracy, as well as reduce the adversarial transferability among base models.

### 3.2 TRS Training

We integrate the proposed TRS regularizer with the standard ensemble training loss, such as Ensemble Cross-Entropy (ECE) loss, to maintain both ensemble model's classification utility and robustness by varying the balancing parameter $\lambda_a$ and $\lambda_b$. Specifically, for an ensemble model consisting of $N$ base models $\{\mathcal{F}_i\}_{i=1}^N$, given an input $(x, y)$, our final training loss train is defined as:

$$\mathcal{L}_{\text{train}} = \frac{1}{N} \sum_{i=1}^N \mathcal{L}_{\text{CE}}(\mathcal{F}_i(x), y) + \frac{2}{N(N-1)} \sum_{i=1}^N \sum_{j=i+1}^N \mathcal{L}_{\text{TRS}}(\mathcal{F}_i, \mathcal{F}_j, x, \delta)$$

where $\mathcal{L}_{\text{CE}}(\mathcal{F}_i(x), y)$ refers to the cross-entropy loss between $\mathcal{F}_i(x)$, the output vector of model $\mathcal{F}_i$ given $x$, and the ground-truth label $y$. The weight of $\mathcal{L}_{\text{TRS}}$ regularizer could be adjusted by the tuning $\lambda_a$ and $\lambda_b$ internally. We present one-epoch training pseudo code in Algorithm 1 of Appendix F. The detailed hyper-parameter setting and training criterion are discussed in Appendix F.

## 4   Experimental Evaluation

In this section, we evaluate the robustness of the proposed TRS-ensemble model under both strong whitebox attacks, as well as blackbox attacks considering the gradient obfuscation concern [1]. We compare TRS with six state-of-the-art ensemble approaches. In addition, we evaluate the adversarial transferability among base models within an ensemble and empirically show that the TRS regularizer can indeed reduce transferability effectively. We also conduct extensive ablation studies to explore the effectiveness of different loss terms in TRS, as well as visualize the trained decision boundaries of different ensemble models to provide intuition on the model properties. We open source the code[3] and provide a large-scale benchmark.

### 4.1   Experimental Setup

**Datasets.** We conduct our experiments on widely-used image datasets including hand-written dataset MNIST [29]; and colourful image datasets CIFAR-10 and CIFAR-100 [26].

---

[3]`https://github.com/AI-secure/Transferability-Reduced-Smooth-Ensemble`

Table 1: Robust accuracy(%) of different ensembles against whitebox attacks on MNIST/CIFAR-10. "para." refers to the attack parameter ($\epsilon$ is the $\ell_\infty$ perturbation budget for the attack and $c$ the constant to balance the attack stealthiness and effectiveness). The first 6 methods are baseline ensembles, and the last 3 columns (Cos-only, Cos-$\ell_2$, TRS) the variants of TRS-ensemble.

| MNIST | para. | AdaBoost | GradientBoost | CKAE | ADP | GAL | DVERGE | Cos-only | Cos-$\ell_2$ | TRS |
|---|---|---|---|---|---|---|---|---|---|---|
| FGSM | $\epsilon = 0.1$ | 70.2 | 73.2 | 72.6 | 71.7 | 35.7 | **95.8** | 66.2 | 91.2 | 95.6 |
| | $\epsilon = 0.2$ | 39.4 | 34.2 | 42.5 | 20.0 | 7.8 | 91.6 | 30.7 | 72.5 | **91.7** |
| BIM (50) | $\epsilon = 0.1$ | 2.6 | 2.4 | 4.2 | 7.7 | 4.6 | 74.9 | 0.4 | 76.2 | **93.3** |
| | $\epsilon = 0.15$ | 0.0 | 0.2 | 0.4 | 0.1 | 2.5 | 47.7 | 0.0 | 47.9 | **85.7** |
| PGD (50) | $\epsilon = 0.1$ | 1.9 | 1.5 | 1.4 | 4.5 | 4.1 | 69.2 | 0.0 | 73.4 | **93.0** |
| | $\epsilon = 0.15$ | 0.0 | 0.0 | 0.5 | 1.0 | 0.6 | 28.8 | 0.0 | 30.2 | **85.1** |
| MIM (50) | $\epsilon = 0.1$ | 1.9 | 1.6 | 1.2 | 13.8 | 0.8 | 75.3 | 0.4 | 74.1 | **92.9** |
| | $\epsilon = 0.15$ | 0.0 | 0.1 | 0.3 | 1.0 | 0.2 | 44.6 | 0.0 | 35.5 | **85.1** |
| CW | $c = 0.1$ | 81.2 | 80.5 | 83.4 | 97.9 | 97.4 | 97.3 | 85.6 | 89.2 | **98.1** |
| | $c = 1.0$ | 66.3 | 65.8 | 69.5 | 90.1 | 68.3 | 79.2 | 58.6 | 54.4 | **92.6** |
| EAD | $c = 5.0$ | 0.2 | 0.1 | 0.1 | 2.2 | 0.2 | 0.0 | 4.1 | 6.9 | **23.3** |
| | $c = 10.0$ | 0.0 | 0.0 | 0.0 | 0.0 | 0.2 | 0.0 | 0.5 | 0.8 | **1.4** |
| APGD-DLR | $\epsilon = 0.1$ | 0.5 | 0.2 | 0.5 | 2.1 | 1.9 | 65.4 | 0.0 | 70.6 | **92.1** |
| | $\epsilon = 0.15$ | 0.0 | 0.0 | 0.1 | 0.5 | 0.2 | 27.4 | 0.0 | 26.3 | **83.4** |
| APGD-CE | $\epsilon = 0.1$ | 0.2 | 0.2 | 0.1 | 1.4 | 1.2 | 63.2 | 0.0 | 69.8 | **91.7** |
| | $\epsilon = 0.15$ | 0.0 | 0.0 | 0.1 | 0.4 | 0.2 | 26.1 | 0.0 | 25.4 | **82.8** |

| CIFAR-10 | para. | AdaBoost | GradientBoost | CKAE | ADP | GAL | DVERGE | Cos-only | Cos-$\ell_2$ | TRS |
|---|---|---|---|---|---|---|---|---|---|---|
| FGSM | $\epsilon = 0.02$ | 28.2 | 30.4 | 34.1 | 58.8 | 19.2 | **63.8** | 56.1 | 35.8 | 44.2 |
| | $\epsilon = 0.04$ | 15.4 | 15.2 | 18.5 | 39.4 | 12.6 | **53.4** | 35.0 | 25.9 | 24.9 |
| BIM (50) | $\epsilon = 0.01$ | 4.2 | 4.4 | 5.1 | 13.8 | 13.0 | 39.1 | 0.0 | 17.1 | **50.6** |
| | $\epsilon = 0.02$ | 0.2 | 0.1 | 0.2 | 0.9 | 2.5 | 13.0 | 0.0 | 1.2 | **15.8** |
| PGD (50) | $\epsilon = 0.01$ | 2.1 | 1.9 | 1.9 | 9.0 | 8.3 | 37.1 | 0.0 | 15.7 | **50.5** |
| | $\epsilon = 0.02$ | 0.0 | 0.0 | 0.2 | 0.1 | 0.6 | 10.5 | 0.0 | 0.5 | **15.1** |
| MIM (50) | $\epsilon = 0.01$ | 2.3 | 1.9 | 2.0 | 18.7 | 10.3 | 40.7 | 0.0 | 18.1 | **51.5** |
| | $\epsilon = 0.02$ | 0.1 | 0.0 | 0.1 | 1.7 | 0.8 | 14.4 | 0.0 | 0.5 | **17.2** |
| CW | $c = 0.01$ | 36.2 | 35.2 | 35.4 | 55.8 | 66.3 | 75.1 | 36.6 | 67.3 | **77.2** |
| | $c = 0.1$ | 18.4 | 26.2 | 23.0 | 25.9 | 28.3 | 57.4 | 17.6 | 30.7 | **58.1** |
| EAD | $c = 1.0$ | 0.2 | 0.0 | 0.0 | 9.0 | 0.0 | 0.2 | 0.0 | 0.0 | **11.7** |
| | $c = 5.0$ | 0.0 | 0.0 | 0.0 | 0.0 | 0.0 | 0.0 | 0.0 | 0.0 | **0.1** |
| APGD-DLR | $\epsilon = 0.01$ | 1.2 | 0.9 | 1.1 | 5.5 | 2.2 | 37.6 | 0.0 | 16.1 | **50.2** |
| | $\epsilon = 0.02$ | 0.0 | 0.0 | 0.0 | 0.2 | 0.0 | 10.2 | 0.0 | 0.5 | **15.1** |
| APGD-CE | $\epsilon = 0.01$ | 0.9 | 0.2 | 0.4 | 3.9 | 1.6 | 37.5 | 0.0 | 15.9 | **48.6** |
| | $\epsilon = 0.02$ | 0.0 | 0.0 | 0.0 | 0.1 | 0.0 | 10.2 | 0.0 | 0.5 | **15.0** |

**Baseline ensemble approaches.** We mainly consider the standard ensemble, as well as the state-of-the-art robust ensemble methods that claim to be resilient against adversarial attacks. Specifically, we consider the following baseline ensemble methods which aim to promote the diversity between base models: **AdaBoost** [19]; **GradientBoost** [16]; **CKAE** [25]; **ADP** [38]; **GAL** [23]; **DVERGE** [56]. The detailed description about these approaches are in Appendix E. DVERGE, which has achieved the state-of-the-art ensemble robustness to our best knowledge, serves as the strongest baseline.

**Whitebox robustness evaluation.** We consider the following adversarial attacks to measure ensembles' whitebox robustness: *Fast Gradient Sign Method* (**FGSM**) [17]; *Basic Iterative Method* (**BIM**) [34]; *Momentum Iterative Method* (**MIM**); *Projected Gradient Descent* (**PGD**); *Auto-PGD* (**APGD**); *Carlini & Wanger Attack* (**CW**); *Elastic-net Attack* (**EAD**) [8], and we leave the detailed description and parameter configuration of these attacks in Appendix E. We use *Robust Accuracy* as our **evaluation metric** for the whitebox setting, defined as the ratio of correctly predicted *adversarial examples* generated by different attacks among the whole test dataset.

**Blackbox robustness evaluation.** We also conduct blackbox robustness analysis in our evaluation since recent studies have shown that robust models which obfuscate gradients could still be fragile under blackbox attacks [1]. In the blackbox attack setting, we assume the attacker has no knowledge about the target ensemble, including the model architecture and parameters. In this case, the attacker is only able to craft adversarial examples based on several surrogate models and transfer them to the target victim ensemble. We follow the same blackbox attack evaluation setting in [56]: We choose three ensembles consisting of $3, 5, 8$ base models which are trained with standard Ensemble Cross-Entropy (ECE) loss as our surrogate models. We apply 50-steps PGD attack with three random starts and two different loss functions (CrossEntropy and CW loss) on each surrogate model to generate adversarial instances (i.e. for each instance we will have 18 attack attempts). For each instance, among these attack attempts, as long as there is one that can successfully attack the victim model, we will count it as a successful attack. In this case, we use *Robust Accuracy* as our **evaluation metric**, defined as the number of unsuccessful attack attempts divided by the number of all attacks. We also consider additional three strong blackbox attacks targeting on reducing transferability (*i.e.*, ILA [21], DI2-SGSM [55], IRA [50]) in Appendix J, which leads to similar observations.

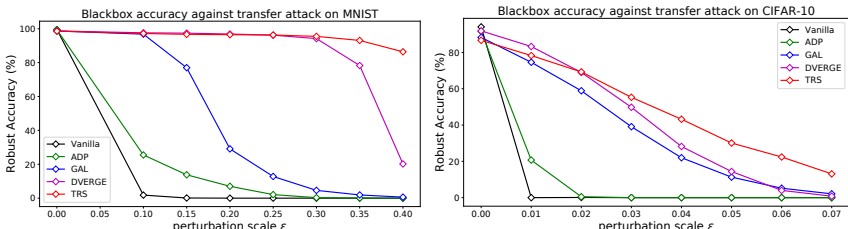

Figure 2: Robust accuracy under blackbox attacks with different $\ell_\infty$ perturbation budget $\epsilon$. (Left): MNIST; (Right): CIFAR-10.

## 4.2 Experimental Results

In this section, we present both whitebox and blackbox robustness evaluation results, examine the adversarial transferability, and explore the impacts of different loss terms in TRS. Furthermore, in Appendix I.1, we visualize the decision boundary; in Appendix I.2, we show results of further improving the robustness of the TRS ensemble by integrating adversarial training; in Appendix I.3, we study the impacts of each of the regularization term $\mathcal{L}_{\text{sim}}$ and $\mathcal{L}_{\text{smooth}}$; in Appendix I.4, we show the convergence of robust accuracy under large attack iterations to demonstrate the robustness stability of TRS ensemble; in Appendix I.5, we analyze the trade-off between the training cost and robustness of TRS by varying PGD step size and the total number of steps within $\mathcal{L}_{\text{smooth}}$ approximation.

**Whitebox robustness.** Table 1 presents the *Robust Accuracy* of different ensembles against a range of whitebox attacks on MNIST and CIFAR-10 dataset. We defer results on CIFAR-100 in Appendix K, and measure the statistical stability of our reported robust accuracy in Appendix H. Results shows that the proposed TRS ensemble outperforms other baselines including the state-of-the-art DVERGE *significantly*, against a range of attacks and perturbation budgets, and such performance gap could be even larger under stronger adversary attacks (e.g. PGD attack). We note that TRS ensemble is slightly less robust than DVERGE under small perturbation with weak attack FGSM. We investigate this based on the decision boundary analysis in Appendix I.1, and find that DVERGE tends to be more robust along the gradient direction and thus more robust against weak attacks which only focus on the gradient direction (e.g., FGSM); while TRS yields a smoother model along different directions leading to more consistent predictions within a larger neighborhood of an input, and thus more robust against strong iterative attacks (e.g., PGD). This may be due to that DVERGE is essentially performing adversarial training for different base models and therefore it protects the adversarial (gradient) direction, while TRS optimizes to train a smooth ensemble with diverse base models. We also analyze the convergence of attack algorithms in Appendix I.4, showing that when the number of attack iterations is large, both ADP and GAL ensemble achieve much lower robust accuracy against such iterative attacks; while both DVERGE and TRS remain robust.

**Blackbox robustness.** Figure 2 shows the *Robust Accuracy* performance of TRS compared with different baseline ensembles under different perturbation budget $\epsilon$. As we can see, the TRS ensemble achieves competitive robust accuracy with DVERGE when $\epsilon$ is very small, and TRS beats *all* the baselines significantly when $\epsilon$ is large. Precisely speaking, TRS ensemble achieves over $85\%$ robust accuracy against transfer attack with $\epsilon = 0.4$ on MNIST while the second-best ensemble (DVERGE) only achieves $20.2\%$. Also on CIFAR-10, TRS ensemble achieves over $25\%$ robust accuracy against transfer attack when $\epsilon = 0.06$, while all the other baseline ensembles achieve robust accuracy lower than $6\%$. This implies that our proposed TRS ensemble has stronger generalization ability in terms of robustness against large $\epsilon$ adversarial attacks compared with other ensembles. We also put more details of the robust accuracy under blackbox attacks in Appendix G.

**Adversarial transferability.** Figure 3 shows the adversarial transferability matrix of different ensembles against 50-steps PGD attack with $\epsilon = 0.3$ for MNIST and $\epsilon = 0.04$ for CIFAR-10. Cell $(i, j)$ where $i \neq j$ represents the transfer attack success rate evaluated on $j$-th base model by using the $i$-th base model as the surrogate model. Lower number in each cell indicates lower transferability and thus potentially higher ensemble robustness. The diagonal cell $(i, i)$ refers to $i$-th base model's attack success rate, which reflects the vulnerability of a single model. From these figures, we can see that while base models show their vulnerabilities against adversarial attack, only DVERGE and TRS ensemble could achieve low adversarial transferability among base models. We should also notice that though GAL applied a similar gradient cosine similarity loss as our loss term $\mathcal{L}_{\text{sim}}$, GAL still can

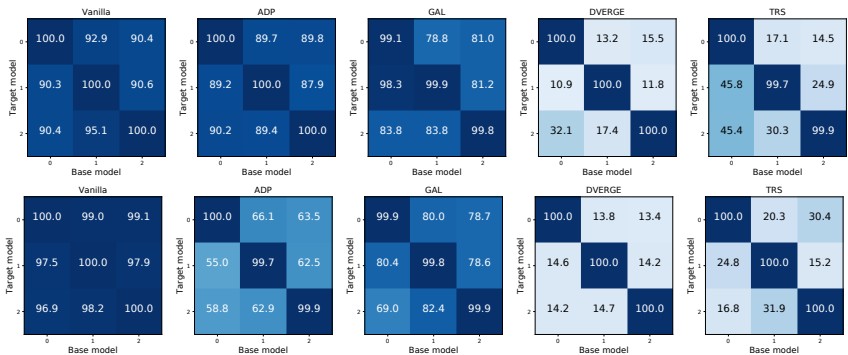

Figure 3: Transferability analysis of PGD attack on MNIST (top) and CIFAR-10 (bottom). Each cell $(i, j)$ shows the *attack success rate* of $i$-th model on the adversarial examples generated against the $j$-th model. We use $\epsilon = 0.3$ for MNIST and $\epsilon = 0.04$ for CIFAR-10.

not achieve low adversarial transferability due to the lack of model smoothness enforcement, which is one of our key contributions in this paper.

**Gradient similarity only vs. TRS.** To further verify our theoretical analysis on the sufficient condition of transferability as model smoothness, we consider only applying *similarity loss* $\mathcal{L}_{\text{sim}}$ without *model smoothness loss* $\mathcal{L}_{\text{smooth}}$ in TRS (i.e. $\lambda_b = 0$). The result is shown as "Cos-only" method of Table 1. We observe that the resulting whitebox robustness is much worse than standard TRS. This matches our theoretical analysis that *only minimizing the gradient similarity cannot guarantee low adversarial transferability among base models* and thus lead to low ensemble robustness. In Appendix I.3, we investigate the impacts of $\mathcal{L}_{\text{sim}}$ and $\mathcal{L}_{\text{smooth}}$ thoroughly, and we show that though $\mathcal{L}_{\text{smooth}}$ contribute slightly more, both terms are critical to the final ensemble robustness.

$\ell_2$ **regularizer only vs. Min-max model smoothing.** To emphasis the importance of our proposed min-max training loss on promoting the margin-wise model smoothness, we train a variant of TRS ensemble Cos-$\ell_2$, where we directly apply the $\ell_2$ regularization on $\|\nabla_x \ell_{\mathcal{F}}\|_2$ and $\|\nabla_x \ell_{\mathcal{G}}\|_2$. The results are shown as "Cos-$\ell_2$" in Table 1. We observe that Cos-$\ell_2$ achieves lower robustness accuracy compared with TRS, which implies the necessity of regularizing the gradient magnitude on not only the local training points but also their neighborhood regions to ensure overall model smoothness.

## 5 Conclusion

In this paper, we deliver an in-depth understanding of adversarial transferability. Theoretically, we provide both lower and upper bounds on transferability which shows that *smooth* models together with *low loss gradient similarity* guarantee low transferability. Inspired by our analysis, we propose TRS ensemble training to empirically reduce transferability by reducing loss gradient similarity and promoting model smoothness, yielding a significant improvement on ensemble robustness.

## Acknowledgments and Disclosure of Funding

This work is partially supported by the NSF grant No.1910100, NSF CNS 20-46726 CAR, the Amazon Research Award, and the joint CATCH MURI-AUSMURI.

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
