In Appendix A, we provide the lower and upper bounds of attack transferability for untargeted attacks. In Appendix B, we extend our theoretical analysis to adversarial attacks bounded by distributional distance. In Appendices C and D, we give the detailed proof of Theorems 1, 2, 3 and 4 characterizing the lower and upper bounds for both targeted and untargeted attack transferability. In Appendix E, we give the detailed introduction of our baseline ensembles and evaluated whitebox attacks, including their exact configuration. In Appendix F, we present all the training details for TRS ensemble and other baselines. In Appendix G, we give the numerical blackbox robustness evaluation results on MNIST and CIFAR-10, corresponding to Figure 2 in main paper. In Appendix H, we analyze the statistical stability of reported robust accuracy for TRS ensemble against attacks with random start, and TRS ensemble claims its stability by showing small standard deviation. In Appendix I, we visualize the decision boundaries of different robust ensembles and investigate how adversarial training would further improve the robustness of TRS ensemble. We also show TRS ensemble remains robust under large attack iterations through convergence analysis. In Appendix J, we evaluate the robustness of TRS ensemble against other three strong blackbox attacks, and TRS ensemble still remains robust. In Appendix K, we conduct whitebox robustness evaluation on CIFAR-100 dataset and compare other state-of-the-art robust ensembles with our proposed TRS ensemble.

# A    Additional Theoretical Results for Untargeted Attacks

In this appendix, we present transferability lower and upper bounds for untargeted attack. All these bounds have similar forms as their targeted attack counterparts in the main text.

## A.1    Lower Bound

**Theorem 3** (Lower Bound on Untargeted Attack Transferability). *Assume both models $\mathcal{F}$ and $\mathcal{G}$ are $\beta$-smooth. Let $\mathcal{A}_U$ be an $(\alpha, \mathcal{F})$-effective untargeted attack with perturbation ball $\|\delta\|_2 \leq \epsilon$. The transferabiity can be lower bounded by*

$$\Pr\left(T_r(\mathcal{F}, \mathcal{G}, x) = 1\right) \geq (1-\alpha) - (\eta_\mathcal{F} + \eta_\mathcal{G}) - \frac{\epsilon(1+\alpha) - c_\mathcal{F}(1-\alpha)}{\epsilon - c_\mathcal{G}} - \frac{\epsilon(1-\alpha)}{\epsilon - c_\mathcal{G}}\sqrt{2 - 2\underline{\mathcal{S}}(\ell_\mathcal{F}, \ell_\mathcal{G})},$$

*where*

$$c_\mathcal{F} = \min_{(x,y)\in\mathrm{supp}(\mathcal{D})} \frac{\min_{y'\in\mathcal{Y}:y'\neq y}\ell_\mathcal{F}(\mathcal{A}_U(x), y') - \ell_\mathcal{F}(x, y) - \beta\epsilon^2/2}{\|\nabla_x\ell_\mathcal{F}(x, y)\|_2}, c_\mathcal{G} = \max_{(x,y)\in\mathrm{supp}(\mathcal{D})} \frac{\min_{y'\in\mathcal{Y}:y'\neq y}\ell_\mathcal{G}(\mathcal{A}_U(x), y') - \ell_\mathcal{G}(x, y) + \beta\epsilon^2/2}{\|\nabla_x\ell_\mathcal{G}(x, y)\|_2}.$$

*Here $\eta_\mathcal{F}$ and $\eta_\mathcal{G}$ are the* risks *of models $\mathcal{F}$ and $\mathcal{G}$ respectively. The* $\mathrm{supp}(\mathcal{D})$ *is the support of benign data distribution, i.e., $x$ is the benign data and $y$ is its associated true label.*

The full proof is available in Appendix C. The discussion of the theorem is in Section 2.

## A.2    Upper Bound

**Theorem 4** (Upper Bound on Untargeted Attack Transferability). *Assume both models $\mathcal{F}$ and $\mathcal{G}$ are $\beta$-smooth with gradient magnitude bounded by $B$, i.e., $\|\nabla_x\ell_\mathcal{F}(x, y)\| \leq B$ and $\|\nabla_x\ell_\mathcal{G}(x, y)\| \leq B$ for any $x \in \mathcal{X}$, $y \in \mathcal{Y}$. Let $\mathcal{A}_U$ be an $(\alpha, \mathcal{F})$-effective untargeted attack with perturbation ball $\|\delta\|_2 \leq \epsilon$. When the attack radius $\epsilon$ is small such that $\ell_{\min} - \epsilon B\left(1 + \sqrt{\frac{1+\overline{\mathcal{S}}(\ell_\mathcal{F}, \ell_\mathcal{G})}{2}}\right) - \beta\epsilon^2 > 0$, the transferability can be upper bounded by*

$$\Pr\left(T_r(\mathcal{F}, \mathcal{G}, x) = 1\right) \leq \frac{\xi_\mathcal{F} + \xi_\mathcal{G}}{\ell_{\min} - \epsilon B\left(1 + \sqrt{\frac{1+\overline{\mathcal{S}}(\ell_\mathcal{F}, \ell_\mathcal{G})}{2}}\right) - \beta\epsilon^2},$$

*where $\ell_{\min} = \min\limits_{\substack{x\in\mathcal{X}, y'\in\mathcal{Y}: \\ (x,y)\in\mathrm{supp}(\mathcal{D}), y'\neq y}} (\ell_\mathcal{F}(x, y'), \ell_\mathcal{G}(x, y'))$. Here $\xi_\mathcal{F}$ and $\xi_\mathcal{G}$ are the* empirical risks *of models $\mathcal{F}$ and $\mathcal{G}$ respectively, defined relative to a differentiable loss. The* $\mathrm{supp}(\mathcal{D})$ *is the support of benign data distribution, i.e., $x$ is the benign data and $y$ is its associated true label.*

The full proof is available in Appendix D. The discussion of the theorem is in Section 2.

# B Discussion: Beyond $\ell_p$ Attack

Besides the widely used $\ell_p$ norm based adversarial examples, here we plan to extend our understanding to the distribution distance analysis.

We no longer distinguish the targeted attack and untargeted attack. Therefore, we denote either of them by $\mathcal{A}$. Accordingly, we revise the definition of $(\alpha, \mathcal{F})$-effective attack (Definition 2) to be $\Pr(\mathcal{F}(\mathcal{A}(x) \neq y)) \geq 1 - \alpha$ where $y \in \mathcal{Y}$ is the true label of $x \in \mathcal{X}$.

Moreover, we use $\mathcal{P}_{\mathcal{A}(x)}$ to represent the distribution of $\mathcal{A}(x) \in \mathcal{X}$ where $x$ is distributed according to $\mathcal{P}_{\mathcal{X}}$.

Now we define the distribution distance that we use to measure the adversarial distribution gap.

**Definition 7** (Total variation distance; [7])**.** *For two probability distributions $\mathcal{P}_{\mathcal{X}}$ and $\mathcal{P}_{\mathcal{A}(x)}$ on $\mathcal{X}$, the total variation distance between them is defined as*

$$\|\mathcal{P}_{\mathcal{X}} - \mathcal{P}_{\mathcal{A}(x)}\|_{TV} = \sup_{C \subseteq \mathcal{X}} |\mathcal{P}_{\mathcal{X}}(C) - \mathcal{P}_{\mathcal{A}(x)}(C)|.$$

Informally, the total variation distance measures the largest change in probability over all events. For discrete probability distributions, the total variation distance is the $\ell_1$ distance between the vectors in the probability simplex representing the two distributions.

**Definition 8.** *Given $\rho \in (0, 1)$, an attack strategy $\mathcal{A}(\cdot)$ is called $\rho$-conservative, if for $x \sim \mathcal{P}_{\mathcal{X}}$, $\|\mathcal{P}_{\mathcal{X}} - \mathcal{P}_{\mathcal{A}(x)}\|_{TV} \leq \rho$.*

This definition formalizes the general objective of generating adversarial examples against deep neural networks: attack samples are likely to be observed, while they do not themselves arouse suspicion.

**Lemma 5.** *Let $f, g : \mathcal{X} \to \mathcal{Y}$ be classifiers, $\delta, \rho, \epsilon \in (0, 1)$ be constants, and $\mathcal{A}(\cdot)$ be an attack strategy. Suppose that $\mathcal{A}(\cdot)$ is $\rho$-conservative and $f, g$ have risk at most $\epsilon$. Then*

$$\Pr(\mathcal{F}(\mathcal{A}(x)) \neq \mathcal{G}(\mathcal{A}(x))) \leq 2\epsilon + \rho$$

*for a given random instance $x \sim \mathcal{P}_{\mathcal{X}}$.*

*Remark.* This result provides theoretical backing for the intuition that the boundaries of low risk classifiers under certain dense data distribution are close [48]. It considers two classifiers that have risk at most $\epsilon$, which indicates their boundaries are close for benign data. It then shows that their boundaries are also close for the perturbed data as long as the attack strategy satisfies a conservative condition which constrains the drift in distribution between the benign and adversarial data.

*Proof of Lemma 5.* Given $\mathcal{A}(\cdot)$ is $\rho$-conservative, by Definition 8 we know

$$|\Pr_{\mathcal{P}_{\mathcal{X}}}(f(\mathcal{A}(x)) = g(\mathcal{A}(x))) - \Pr_{\mathcal{X}}(f(x) = g(x))|$$
$$= |\Pr_{\mathcal{A}(x)}(f(x) = g(x)) - \Pr_{\mathcal{X}}(f(x) = g(x))|$$
$$\leq \rho.$$

Therefore, we have

$$\Pr(f(\mathcal{A}(x)) = g(\mathcal{A}(x))) \geq \Pr(f(x) = g(x)) - \rho.$$

From the low-risk conditions, the classifiers agree with high probability:

$$\Pr(f(\mathcal{A}(x)) \neq g(\mathcal{A}(x)))$$
$$\leq \Pr(f(x) \neq g(x)) + \rho$$
$$\leq 1 - \Pr(f(x) = y, g(x) = y) + \rho, \quad [4]$$
$$\leq 1 - (1 - \Pr(f(x) \neq y) - \Pr(g(x) \neq y)) + \rho$$
$$= \epsilon + \epsilon + \rho$$
$$\leq 2\epsilon + \rho,$$

where the third inequality follows from the union bound. [5] $\qquad \square$

---

[4] Here we assume $y$ is the ground truth label.

[5] Recall that for arbitrary events $A_1, \ldots, A_n$, the union bound implies $P\left(\bigcap_{i=1}^n A_i\right) \geq 1 - \sum_{i=1}^n P\left(\overline{A_i}\right)$.

**Theorem 6.** *Let $\mathcal{F}, \mathcal{G} : \mathcal{X} \to \mathcal{Y}$ be classifiers ($\mathcal{Y} \in \{-1, 1\}$), $\delta, \rho, \epsilon \in (0, 1)$ be constants, and $\mathcal{A}(\cdot)$ an attack strategy. Suppose that $\mathcal{A}(\cdot)$ is $\rho$-conservative and $\mathcal{F}, \mathcal{G}$ have risk at most $\epsilon$. Given random instance $x \in \mathcal{X}$, if $\mathcal{A}(\cdot)$ is $(\delta, \mathcal{F})$-effective, then it is also $(\delta + 2\epsilon + \rho, \mathcal{G})$-effective.*

The proof is shown below. This result formalizes the intuition that low-risk classifiers possess close decision boundaries in high-probability regions. In such settings, an attack strategy that successfully attacks one classifier would have high probability to mislead the other. This theorem explains why we should expect successful transferability in practice: defenders will naturally prefer low-risk binary classifiers. This desirable quality of classifiers is a potential liability.

*Proof of Theorem 6.* From Lemma 5 and the union bound we have

$$
\begin{aligned}
&\Pr\left(g(x) \neq y\right) \\
\geq & \Pr\left(f(\mathcal{A}(x)) \neq y, g(\mathcal{A}(x)) = f(\mathcal{A}(x))\right) \\
\geq & 1 - \Pr\left(f(\mathcal{A}(x)) = y\right) - \Pr\left(g(\mathcal{A}(x)) \neq f(\mathcal{A}(x))\right) \\
\geq & 1 - \delta - 2\epsilon - \rho,
\end{aligned}
$$

as claimed. $\square$

## C   Proof of Transferability Lower Bound (Theorems 1 and 3)

Here we present the proof of Theorem 1 and Theorem 3 stated in Section 2.3 and Appendix A.

The following lemma is used in the proof.

**Lemma 7.** *For arbitrary vector $\delta$, $x$, $y$, suppose $\|\delta\|_2 \leq \epsilon$, $x$ and $y$ are unit vectors, i.e., $\|x\|_2 = \|y\|_2 = 1$. Let $m := \cos\langle x, y \rangle = \dfrac{x \cdot y}{\|x\|_2 \cdot \|y\|_2}$. Let $c$ denote any real number. Then*

$$
\delta \cdot y > c + \epsilon\sqrt{2 - 2m} \quad \Rightarrow \quad \delta \cdot x > c.
$$

*Proof.* $\delta \cdot x = \delta \cdot y + \delta \cdot (x - y) > c + \epsilon\sqrt{2 - 2m} + \delta \cdot (x - y)$. By law of cosines, $\delta \cdot (x - y) \geq -\epsilon\sqrt{2 - 2\cos\langle x, y \rangle} = -\epsilon\sqrt{2 - 2m}$. Hence, $\delta \cdot x > c$. $\square$

**Theorem** (Lower Bound on Targeted Attack Transferability)**.** *Assume both models $\mathcal{F}$ and $\mathcal{G}$ are $\beta$-smooth. Let $\mathcal{A}_T$ be an $(\alpha, \mathcal{F})$-effective targeted attack with perturbation ball $\|\delta\|_2 \leq \epsilon$ and target label $y_t \in \mathcal{Y}$. The transferabiity can be lower bounded by*

$$
\Pr\left(T_r(\mathcal{F}, \mathcal{G}, x, y_t) = 1\right) \geq (1 - \alpha) - (\eta_{\mathcal{F}} + \eta_{\mathcal{G}}) - \frac{\epsilon(1 + \alpha) + c_{\mathcal{F}}(1 - \alpha)}{c_{\mathcal{G}} + \epsilon} - \frac{\epsilon(1 - \alpha)}{c_{\mathcal{G}} + \epsilon}\sqrt{2 - 2\underline{\mathcal{S}}(\ell_{\mathcal{F}}, \ell_{\mathcal{G}})},
$$

*where*

$$
c_{\mathcal{F}} = \max_{x \in \mathcal{X}} \frac{\min\limits_{y \in \mathcal{Y}} \ell_{\mathcal{F}}(\mathcal{A}_T(x), y) - \ell_{\mathcal{F}}(x, y_t) + \beta\epsilon^2/2}{\|\nabla_x \ell_{\mathcal{F}}(x, y_t)\|_2},
$$

$$
c_{\mathcal{G}} = \min_{x \in \mathcal{X}} \frac{\min\limits_{y \in \mathcal{Y}} \ell_{\mathcal{G}}(\mathcal{A}_T(x), y) - \ell_{\mathcal{G}}(x, y_t) - \beta\epsilon^2/2}{\|\nabla_x \ell_{\mathcal{G}}(x, y_t)\|_2}.
$$

*Here $\eta_{\mathcal{F}}, \eta_{\mathcal{G}}$ are the risks of models $\mathcal{F}$ and $\mathcal{G}$ respectively.*

*Proof.* For simplifying the notations, we define $x^{\mathcal{A}} := \mathcal{A}_T(x)$, which is the generated adversarial example by $\mathcal{A}_T$ when the input is $x$.

Define auxiliary function $f, g : \mathcal{X} \mapsto \mathbb{R}$ such that

$$
f(x) = \frac{\min_{y \in \mathcal{Y}} \ell_{\mathcal{F}}(x^{\mathcal{A}}, y) - \ell_{\mathcal{F}}(x, y_t) + \beta\epsilon^2/2}{\|\nabla_x \ell_{\mathcal{F}}(x, y_t)\|_2},
$$

$$
g(x) = \frac{\min_{y \in \mathcal{Y}} \ell_{\mathcal{G}}(x^{\mathcal{A}}, y) - \ell_{\mathcal{G}}(x, y_t) - \beta\epsilon^2/2}{\|\nabla_x \ell_{\mathcal{G}}(x, y_t)\|_2}.
$$

The $f$ and $g$ are orthogonal to the confidence score functions of model $\mathcal{F}$ and $\mathcal{G}$. Note that $c_{\mathcal{F}} = \max_{x \in \mathcal{X}} f(x)$ and $c_{\mathcal{G}} = \min_{x \in \mathcal{X}} g(x)$.

The transferability of concern satisfies:

$$
\begin{aligned}
&\Pr\left(T_r(\mathcal{F},\mathcal{G},x,y_t)=1\right) \\
=&\Pr\left(\mathcal{F}(x)=y \cap \mathcal{G}(x)=y \cap \mathcal{F}(x^{\mathcal{A}})=y_t \cap \mathcal{G}(x^{\mathcal{A}})=y_t\right) \quad &(2)\\
\geq& 1 - \Pr\left(\mathcal{F}(x)\neq y\right) - \Pr\left(\mathcal{G}(x)\neq y\right) - \Pr\left(\mathcal{F}(x^{\mathcal{A}})\neq y_t\right) - \Pr\left(\mathcal{G}(x^{\mathcal{A}})\neq y_t\right) \quad &(3)\\
\geq& 1 - \eta_{\mathcal{F}} - \eta_{\mathcal{G}} - \alpha - \Pr\left(\mathcal{G}(x^{\mathcal{A}})\neq y_t\right). \quad &(4)
\end{aligned}
$$

Eq. 2 follows the definition (Definition 6). Eq. 2 to Eq. 3 follows from the union bound. From Eq. 2 to Eq. 3 definition of model risk (Definition 3) and definition of adversarial effectiveness (Definition 2) are applied.

Now consider $\Pr\left(\mathcal{F}(x^{\mathcal{A}})\neq y_t\right)$ and $\Pr\left(\mathcal{G}(x^{\mathcal{A}})\neq y_t\right)$. Given that model predicts the label for which $\ell_{\mathcal{F}}$ is minimized, $\mathcal{F}(x^{\mathcal{A}}) \neq y_t \iff \ell_{\mathcal{F}}(x+\delta,y_t) > \min_y \ell_{\mathcal{F}}(x+\delta,y)$. Similarly, $\mathcal{G}(x^{\mathcal{A}}) \neq y_t \iff \ell_{\mathcal{G}}(x+\delta,y_t) > \min_y \ell_{\mathcal{G}}(x+\delta,y)$.

Following Taylor's Theorem with Lagrange remainder, we have

$$
\ell_{\mathcal{F}}(x+\delta,y_t) = \ell_{\mathcal{F}}(x,y_t) + \delta\nabla_x\ell_{\mathcal{F}}(x,y_t) + \frac{1}{2}\xi^{\top}\mathbf{H}_{\mathcal{F}}\xi, \quad (5)
$$

$$
\ell_{\mathcal{G}}(x+\delta,y_t) = \ell_{\mathcal{G}}(x,y_t) + \delta\nabla_x\ell_{\mathcal{G}}(x,y_t) + \frac{1}{2}\xi^{\top}\mathbf{H}_{\mathcal{G}}\xi. \quad (6)
$$

In Eq. 5 and Eq. 6, $\xi = k\delta$ for some $k \in [0,1]$. $\mathbf{H}_{\mathcal{F}}$ and $\mathbf{H}_{\mathcal{G}}$ are Hessian matrices of $\ell_{\mathcal{F}}$ and $\ell_{\mathcal{G}}$ respectively. Since $\ell_{\mathcal{F}}(x+\delta,y_t)$ and $\ell_{\mathcal{G}}(x+\delta,y_t)$ are $\beta$-smooth, the maximum eigenvalues of $\mathbf{H}_{\mathcal{F}}$ and $\mathbf{H}_{\mathcal{G}}$ are bounded by $\beta$, As the result, $|\xi^{\top}\mathbf{H}_{\mathcal{F}}\xi| \leq \beta \cdot \|\xi\|_2^2 \leq \beta\epsilon^2$. Applying them to Eq. 5 and Eq. 6, we thus have

$$
\ell_{\mathcal{F}}(x,y_t) + \delta\nabla_x\ell_{\mathcal{F}}(x,y_t) - \frac{1}{2}\beta\epsilon^2 \leq \ell_{\mathcal{F}}(x+\delta,y_t) \leq \ell_{\mathcal{F}}(x,y_t) + \delta\nabla_x\ell_{\mathcal{F}}(x,y_t) + \frac{1}{2}\beta\epsilon^2, \quad (7)
$$

$$
\ell_{\mathcal{G}}(x,y_t) + \delta\nabla_x\ell_{\mathcal{G}}(x,y_t) - \frac{1}{2}\beta\epsilon^2 \leq \ell_{\mathcal{G}}(x+\delta,y_t) \leq \ell_{\mathcal{G}}(x,y_t) + \delta\nabla_x\ell_{\mathcal{G}}(x,y_t) + \frac{1}{2}\beta\epsilon^2. \quad (8)
$$

Apply left hand side of Eq. 7 to $\Pr\left(\mathcal{F}(x^{\mathcal{A}})\neq y_t\right) \leq \alpha$ (from Definition 2):

$$
\begin{aligned}
&\Pr\left(\mathcal{F}(x^{\mathcal{A}})\neq y_t\right) \\
=&\Pr\left(\ell_{\mathcal{F}}(x+\delta,y_t) > \min_y \ell_{\mathcal{F}}(x+\delta,y)\right) \\
\geq&\Pr\left(\ell_{\mathcal{F}}(x,y_t) + \delta\nabla_x\ell_{\mathcal{F}}(x,y_t) - \frac{1}{2}\beta\epsilon^2 > \min_y \ell_{\mathcal{F}}(x+\delta,y)\right) \\
=&\Pr\left(\delta \cdot \frac{\nabla_x\ell_{\mathcal{F}}(x,y_t)}{\|\nabla_x\ell_{\mathcal{F}}(x,y_t)\|_2} > f(x)\right), \\
\Longrightarrow& \Pr\left(\delta \cdot \frac{\nabla_x\ell_{\mathcal{F}}(x,y_t)}{\|\nabla_x\ell_{\mathcal{F}}(x,y_t)\|_2} > f(x)\right) \leq \alpha.
\end{aligned}
$$

Similarly, we apply right hand side of Eq. 8 to $\Pr\left(\mathcal{G}(x^{\mathcal{A}})=y_t\right)$:

$$
\begin{aligned}
&\Pr\left(\mathcal{G}(x^{\mathcal{A}})\neq y_t\right) \\
=&\Pr\left(\ell_{\mathcal{G}}(x+\delta,y_t) > \min_y \ell_{\mathcal{G}}(x+\delta,y)\right) \\
\leq&\Pr\left(\ell_{\mathcal{G}}(x,y_t) + \delta\nabla_x\ell_{\mathcal{G}}(x,y_t) + \frac{1}{2}\beta\epsilon^2 > \min_y \ell_{\mathcal{G}}(x+\delta,y)\right) \\
=&\Pr\left(\delta \cdot \frac{\nabla_x\ell_{\mathcal{G}}(x,y_t)}{\|\nabla_x\ell_{\mathcal{G}}(x,y_t)\|_2} > g(x)\right). \quad (9)
\end{aligned}
$$

Knowing that $\|\delta\|_2 \leq \epsilon$, from Lemma 7 we have

$$
\delta \cdot \frac{\nabla_x\ell_{\mathcal{G}}(x,y_t)}{\|\nabla_x\ell_{\mathcal{G}}(x,y_t)\|_2} > f(x) + \epsilon\sqrt{2 - 2\underline{\mathcal{S}}(\ell_{\mathcal{F}},\ell_{\mathcal{G}})} \quad (10)
$$

$$\implies \delta \cdot \frac{\nabla_x \ell_{\mathcal{G}}(x, y_t)}{\|\nabla_x \ell_{\mathcal{G}}(x, y_t)\|_2} > f(x) + \epsilon \sqrt{2 - 2 \cos\langle \nabla_x \ell_{\mathcal{F}}(x, y_t), \nabla_x \ell_{\mathcal{G}}(x, y_t)\rangle} \qquad (11)$$

$$\implies \delta \cdot \frac{\nabla_x \ell_{\mathcal{F}}(x, y_t)}{\|\nabla_x \ell_{\mathcal{F}}(x, y_t)\|_2} > f(x). \qquad (12)$$

From Eq. 10 to Eq. 11, the infimum in definition of $\underline{\mathcal{S}}$ (Definition 4) indicates that

$$\underline{\mathcal{S}}(\ell_{\mathcal{F}}, \ell_{\mathcal{G}}) \leq \cos\langle \nabla_x \ell_{\mathcal{F}}(x, y_t), \nabla_x \ell_{\mathcal{G}}(x, y_t)\rangle.$$

Hence,

$$f(x) + \epsilon \sqrt{2 - 2\underline{\mathcal{S}}(\ell_{\mathcal{F}}, \ell_{\mathcal{G}})} \geq f(x) + \epsilon \sqrt{2 - 2 \cos\langle \nabla_x \ell_{\mathcal{F}}(x, y_t), \nabla_x \ell_{\mathcal{G}}(x, y_t)\rangle}.$$

Eq. 11 to Eq. 12 directly uses Lemma 7. As a result,

$$\Pr\left( \delta \cdot \frac{\nabla_x \ell_{\mathcal{G}}(x, y_t)}{\|\nabla_x \ell_{\mathcal{G}}(x, y_t)\|_2} > f(x) + \epsilon \sqrt{2 - 2\underline{\mathcal{S}}(\ell_{\mathcal{F}}, \ell_{\mathcal{G}})} \right)$$

$$\leq \Pr\left( \delta \cdot \frac{\nabla_x \ell_{\mathcal{F}}(x, y_t)}{\|\nabla_x \ell_{\mathcal{F}}(x, y_t)\|_2} > f(x) \right) \leq \alpha.$$

Note that $f(x) \leq c_{\mathcal{F}}$, we have

$$\Pr\left( \delta \cdot \frac{\nabla_x \ell_{\mathcal{G}}(x, y_t)}{\|\nabla_x \ell_{\mathcal{G}}(x, y_t)\|_2} > c_{\mathcal{F}} + \epsilon \sqrt{2 - 2\underline{\mathcal{S}}(\ell_{\mathcal{F}}, \ell_{\mathcal{G}})} \right) \leq \alpha.$$

Now we consider the maximum expectation of $\delta \cdot \frac{\nabla_x \ell_{\mathcal{G}}(x, y_t)}{\|\nabla_x \ell_{\mathcal{G}}(x, y_t)\|_2}$. Its maximum is $\max \|\delta\|_2 = \epsilon$. Therefore, its expectation is bounded:

$$\mathbb{E}\left[ \delta \cdot \frac{\nabla_x \ell_{\mathcal{G}}(x, y_t)}{\|\nabla_x \ell_{\mathcal{G}}(x, y_t)\|_2} \right] \leq \epsilon \cdot \alpha + \left( c_{\mathcal{F}} + \epsilon \sqrt{2 - 2\underline{\mathcal{S}}(\ell_{\mathcal{F}}, \ell_{\mathcal{G}})} \right)(1 - \alpha).$$

Now applying Markov's inequality, we get

$$\Pr\left( \delta \cdot \frac{\nabla_x \ell_{\mathcal{G}}(x, y_t)}{\|\nabla_x \ell_{\mathcal{G}}(x, y_t)\|_2} > c_{\mathcal{G}} \right)$$

$$\leq \frac{\epsilon \cdot \alpha + \left( c_{\mathcal{F}} + \epsilon \sqrt{2 - 2\underline{\mathcal{S}}(\ell_{\mathcal{F}}, \ell_{\mathcal{G}})} \right)(1 - \alpha) + \epsilon}{c_{\mathcal{G}} + \epsilon}$$

$$= \frac{\epsilon(1 + \alpha) + \left( c_{\mathcal{F}} + \epsilon \sqrt{2 - 2\underline{\mathcal{S}}(\ell_{\mathcal{F}}, \ell_{\mathcal{G}})} \right)(1 - \alpha)}{c_{\mathcal{G}} + \epsilon}.$$

Since $g(x) \geq c_{\mathcal{G}}$,

$$\Pr\left( \delta \cdot \frac{\nabla_x \ell_{\mathcal{G}}(x, y_t)}{\|\nabla_x \ell_{\mathcal{G}}(x, y_t)\|_2} > g(x) \right) \leq \Pr\left( \delta \cdot \frac{\nabla_x \ell_{\mathcal{G}}(x, y_t)}{\|\nabla_x \ell_{\mathcal{G}}(x, y_t)\|_2} > c_{\mathcal{G}} \right)$$

$$\leq \frac{\epsilon(1 + \alpha) + \left( c_{\mathcal{F}} + \epsilon \sqrt{2 - 2\underline{\mathcal{S}}(\ell_{\mathcal{F}}, \ell_{\mathcal{G}})} \right)(1 - \alpha)}{c_{\mathcal{G}} + \epsilon}.$$

Combine with Eq. 12, finally,

$$\Pr\left( T_r(\mathcal{F}, \mathcal{G}, x, y_t) = 1 \right)$$

$$\geq 1 - \eta_{\mathcal{F}} - \eta_{\mathcal{G}} - \alpha - \Pr\left( \mathcal{G}(x^{\mathcal{A}}) \neq y_t \right)$$

$$\overset{(i.)}{\geq} 1 - \eta_{\mathcal{F}} - \eta_{\mathcal{G}} - \alpha - \Pr\left( \delta \cdot \frac{\nabla_x \ell_{\mathcal{G}}(x, y_t)}{\|\nabla_x \ell_{\mathcal{G}}(x, y_t)\|_2} > g(x) \right)$$

$$\geq 1 - \eta_{\mathcal{F}} - \eta_{\mathcal{G}} - \alpha - \frac{\epsilon(1 + \alpha) + \left( c_{\mathcal{F}} + \epsilon \sqrt{2 - 2\underline{\mathcal{S}}(\ell_{\mathcal{F}}, \ell_{\mathcal{G}})} \right)(1 - \alpha)}{c_{\mathcal{G}} + \epsilon}$$

$$= (1 - \alpha) - (\eta_{\mathcal{F}} + \eta_{\mathcal{G}}) - \frac{\epsilon(1 + \alpha) + c_{\mathcal{F}}(1 - \alpha)}{c_{\mathcal{G}} + \epsilon} - \frac{\epsilon(1 - \alpha)}{c_{\mathcal{G}} + \epsilon} \sqrt{2 - 2\underline{\mathcal{S}}(\ell_{\mathcal{F}}, \ell_{\mathcal{G}})}.$$

Here, $(i.)$ follows Eq. 9. $\qquad \square$

**Theorem** (Lower Bound on Untargeted Attack Transferability). *Assume both models $\mathcal{F}$ and $\mathcal{G}$ are $\beta$-smooth. Let $\mathcal{A}_U$ be an $(\alpha, \mathcal{F})$-effective untargeted attack with perturbation ball $\|\delta\|_2 \leq \epsilon$. The transferabiity can be lower bounded by*

$$\Pr\left(T_r(\mathcal{F}, \mathcal{G}, x) = 1\right) \geq (1-\alpha) - (\eta_\mathcal{F} + \eta_\mathcal{G}) - \frac{\epsilon(1+\alpha) - c_\mathcal{F}(1-\alpha)}{\epsilon - c_\mathcal{G}} - \frac{\epsilon(1-\alpha)}{\epsilon - c_\mathcal{G}}\sqrt{2 - 2\underline{\mathcal{S}}(\ell_\mathcal{F}, \ell_\mathcal{G})},$$

*where*

$$c_\mathcal{F} = \min_{(x,y)\in\mathrm{supp}(\mathcal{D})} \frac{\min\limits_{y'\in\mathcal{Y}:y'\neq y} \ell_\mathcal{F}(\mathcal{A}_U(x), y') - \ell_\mathcal{F}(x,y) - \beta\epsilon^2/2}{\|\nabla_x \ell_\mathcal{F}(x,y)\|_2},$$

$$c_\mathcal{G} = \max_{(x,y)\in\mathrm{supp}(\mathcal{D})} \frac{\min\limits_{y'\in\mathcal{Y}:y'\neq y} \ell_\mathcal{G}(\mathcal{A}_U(x), y') - \ell_\mathcal{G}(x,y) + \beta\epsilon^2/2}{\|\nabla_x \ell_\mathcal{G}(x,y)\|_2}.$$

*Here $\eta_\mathcal{F}$ and $\eta_\mathcal{G}$ are the* risks *of models $\mathcal{F}$ and $\mathcal{G}$ respectively. The $\mathrm{supp}(\mathcal{D})$ is the support of benign data distribution, i.e., $x$ is the benign data and $y$ is its associated true label.*

*Proof.* For simplifying the notations, we define $x^\mathcal{A} := \mathcal{A}_U(x)$, which is the generated adversarial example by $\mathcal{A}_U$ when the input is $x$. Define auxiliary function $f, g : \mathcal{M} \to \mathbb{R}$ such that

$$f(x,y) = \frac{\min\limits_{y'\in\mathcal{Y}:y'\neq y} \ell_\mathcal{F}(x^\mathcal{A}, y') - \ell_\mathcal{F}(x,y) - \beta\epsilon^2/2}{\|\nabla_x \ell_\mathcal{F}(x, y_t)\|_2},$$

$$g(x,y) = \frac{\min\limits_{y'\in\mathcal{Y}:y'\neq y} \ell_\mathcal{G}(x^\mathcal{A}, y') - \ell_\mathcal{G}(x,y) + \beta\epsilon^2/2}{\|\nabla_x \ell_\mathcal{G}(x, y_t)\|_2}.$$

The $f$ and $g$ are orthogonal to the confidence score functions of model $\mathcal{F}$ and $\mathcal{G}$. Note that

$$c_\mathcal{F} = \min_{(x,y)\in\mathrm{supp}(\mathcal{D})} f(x,y), c_\mathcal{G} = \max_{(x,y)\in\mathrm{supp}(\mathcal{D})} g(x,y).$$

The proof is similar to that of Theorem 1.

$$\begin{aligned}
&\Pr\left(T_r(\mathcal{F}, \mathcal{G}, x) = 1\right) \\
=&\Pr\left(\mathcal{F}(x) = y \cap \mathcal{G}(x) = y \cap \mathcal{F}(x^\mathcal{A}) \neq y \cap \mathcal{G}(x^\mathcal{A}) \neq y\right) \\
\geq&1 - \Pr\left(\mathcal{F}(x) \neq y\right) - \Pr\left(\mathcal{G}(x) \neq y\right) - \Pr\left(\mathcal{F}(x^\mathcal{A}) = y\right) - \Pr\left(\mathcal{G}(x^\mathcal{A}) = y\right) \\
=&1 - \eta_\mathcal{F} - \eta_\mathcal{G} - \alpha - \Pr\left(\mathcal{G}(x^\mathcal{A}) = y\right).
\end{aligned} \tag{13}$$

From Taylor's Theorem and Lemma 7, we observe that

$$\Pr\left(\mathcal{G}(x^\mathcal{A}) = y\right) \leq \Pr\left(\delta \cdot \frac{\nabla_x \ell_\mathcal{G}(x,y)}{\|\nabla_x \ell_\mathcal{G}(x,y)\|_2} < c_\mathcal{G}\right), \tag{14}$$

$$\Pr\left(\delta \cdot \frac{\nabla_x \ell_\mathcal{G}(x,y)}{\|\nabla_x \ell_\mathcal{G}(x,y)\|_2} < c_\mathcal{F} - \epsilon\sqrt{2 - 2\underline{\mathcal{S}}(\ell_\mathcal{F}, \ell_\mathcal{G})}\right) \leq \Pr\left(\mathcal{F}(x^\mathcal{A}) = y\right) = \alpha. \tag{15}$$

According to Markov's inequality, Eq. 15 implies that

$$\Pr\left(\delta \cdot \frac{\nabla_x \ell_\mathcal{G}(x,y)}{\|\nabla_x \ell_\mathcal{G}(x,y)\|_2} < c_\mathcal{G}\right) \leq \frac{\epsilon(1+\alpha) - \left(c_\mathcal{F} - \epsilon\sqrt{2 - 2\underline{\mathcal{S}}(\ell_\mathcal{F}, \ell_\mathcal{G})}\right)(1-\alpha)}{\epsilon - c_\mathcal{G}}. \tag{16}$$

We conclude the proof by combining Eq. 14 with Eq. 16 and plugging it into Eq. 13. $\square$

## D  Proof of Transferability Upper Bound (Theorems 2 and 4)

Here we present the proof of Theorem 2 and Theorem 4 as stated in Section 2.4 and Appendix A.

The following lemma is used in the proof.

**Lemma 8.** *Suppose two unit vectors $x, y$ satisfy $x \cdot y \leq S$, then for any $\delta$, we have $\min(\delta \cdot x, \delta \cdot y) \leq \|\delta\|_2\sqrt{(1+S)/2}$.*

*Proof.* Denote $\alpha$ to be the angle between $x$ and $y$, then $\cos\alpha \leq S$, or $\alpha \geq \arccos S$. If $\alpha_x, \alpha_y$ are the angles between $\delta$ and $x$ and between $\delta$ and $y$ respectively, then we have $\max(\alpha_x, \alpha_y) \geq \alpha/2 \geq \arccos S/2$. By the half-angle formula, $\cos(\alpha/2) \leq \cos\left(\frac{\arccos S}{2}\right) = \sqrt{\frac{1+S}{2}}$. Thus, $\min(\delta \cdot x, \delta \cdot y) \leq \|\delta\|_2 \cos(\alpha/2) \leq \|\delta\|_2 \sqrt{(1+S)/2}$. $\qquad\square$

**Theorem** (Upper Bound on Targeted Attack Transferability). *Assume both model $\mathcal{F}$ and $\mathcal{G}$ are $\beta$-smooth with gradient magnitude bounded by $B$, i.e., $\|\nabla_x\ell_{\mathcal{F}}(x,y)\| \leq B$ and $\|\nabla_x\ell_{\mathcal{G}}(x,y)\| \leq B$ for any $x \in \mathcal{X}, y \in \mathcal{Y}$. Let $\mathcal{A}_T$ be an $(\alpha, \mathcal{F})$-effective targeted attack with perturbation ball $\|\delta\|_2 \leq \epsilon$ and target label $y_t \in \mathcal{Y}$. When the attack radius $\epsilon$ is small such that $\ell_{\min} - \epsilon B\left(1 + \sqrt{\frac{1+\overline{\mathcal{S}}(\ell_{\mathcal{F}},\ell_{\mathcal{G}})}{2}}\right) - \beta\epsilon^2 > 0$, the transferability can be upper bounded by*

$$\Pr\left(T_r(\mathcal{F},\mathcal{G},x,y_t) = 1\right) \leq \frac{\xi_{\mathcal{F}} + \xi_{\mathcal{G}}}{\ell_{\min} - \epsilon B\left(1 + \sqrt{\frac{1+\overline{\mathcal{S}}(\ell_{\mathcal{F}},\ell_{\mathcal{G}})}{2}}\right) - \beta\epsilon^2},$$

*where $\ell_{\min} = \min_{x\in\mathcal{X}}\left(\ell_{\mathcal{F}}(x,y_t), \ell_{\mathcal{G}}(x,y_t)\right)$. Here $\xi_{\mathcal{F}}$ and $\xi_{\mathcal{G}}$ are the* empirical risks *of models $\mathcal{F}$ and $\mathcal{G}$ respectively, defined relative to a differentiable loss.*

*Proof.* We let $x^{\mathcal{A}} := \mathcal{A}_T(x)$ be the generated adversarial example when the input is $x$. Since $\mathcal{F}(x)$ outputs label for which $\ell_{\mathcal{F}}$ is minimized, we have

$$\mathcal{F}(x) = y \implies \ell_{\mathcal{F}}(x, y_t) > \ell_{\mathcal{F}}(x, y) \tag{17}$$

and similarly

$$\mathcal{F}(x^{\mathcal{A}}) = y_t \implies \ell_{\mathcal{F}}(x^{\mathcal{A}}, y) > \ell_{\mathcal{F}}(x^{\mathcal{A}}, y_t), \tag{18}$$
$$\mathcal{G}(x) = y \implies \ell_{\mathcal{G}}(x, y_t) > \ell_{\mathcal{G}}(x, y), \tag{19}$$
$$\mathcal{G}(x^{\mathcal{A}}) = y_t \implies \ell_{\mathcal{G}}(x^{\mathcal{A}}, y) > \ell_{\mathcal{G}}(x^{\mathcal{A}}, y_t). \tag{20}$$

Since $\ell_{\mathcal{F}}(x,y)$ and $\ell_{\mathcal{G}}(x,y)$ are $\beta$-smooth,

$$\ell_{\mathcal{F}}(x,y) + \delta \cdot \nabla_x\ell_{\mathcal{F}}(x,y) + \frac{\beta}{2}\|\delta\|^2 \geq \ell_{\mathcal{F}}(x^{\mathcal{A}}, y),$$

which implies

$$\begin{aligned}
\delta \cdot \nabla_x\ell_{\mathcal{F}}(x,y) &\geq \ell_{\mathcal{F}}(x^{\mathcal{A}}, y) - \ell_{\mathcal{F}}(x,y) - \frac{\beta}{2}\|\delta\|^2 \\
&\geq \ell_{\mathcal{F}}(x^{\mathcal{A}}, y_t) - \ell_{\mathcal{F}}(x,y) - \frac{\beta}{2}\|\delta\|^2 =: c'_{\mathcal{F}}.
\end{aligned} \tag{21}$$

Similarly for $\mathcal{G}$,

$$\delta \cdot \nabla_x\ell_{\mathcal{G}}(x,y) \geq \ell_{\mathcal{G}}(x^{\mathcal{A}}, y_t) - \ell_{\mathcal{G}}(x,y) - \frac{\beta}{2}\|\delta\|^2 =: c'_{\mathcal{G}}. \tag{22}$$

Thus,

$$\Pr\left(\mathcal{F}(x) = y, \mathcal{G}(x) = y, \mathcal{F}(x^{\mathcal{A}}) = y_t, \mathcal{G}(x^{\mathcal{A}}) = y_t\right)$$
$$\leq \Pr\left(\ell_{\mathcal{F}}(x,y_t) > \ell_{\mathcal{F}}(x,y), \ell_{\mathcal{F}}(x^{\mathcal{A}},y) > \ell_{\mathcal{F}}(x^{\mathcal{A}},y_t), \ell_{\mathcal{G}}(x,y_t) > \ell_{\mathcal{G}}(x,y), \ell_{\mathcal{G}}(x^{\mathcal{A}},y) > \ell_{\mathcal{G}}(x^{\mathcal{A}},y_t)\right) \tag{23}$$
$$\leq \Pr\left(\delta \cdot \nabla_x\ell_{\mathcal{F}}(x,y) \geq c'_{\mathcal{F}}, \, \delta \cdot \nabla_x\ell_{\mathcal{G}}(x,y) \geq c'_{\mathcal{G}}\right) \tag{24}$$
$$\leq \Pr\left(\left(c'_{\mathcal{F}} \leq \epsilon\sqrt{(1+\overline{\mathcal{S}}(\ell_{\mathcal{F}},\ell_{\mathcal{G}}))/2}\|\nabla_x\ell_{\mathcal{F}}(x,y)\|_2\right) \bigcup \left(c'_{\mathcal{G}} \leq \epsilon\sqrt{(1+\overline{\mathcal{S}}(\ell_{\mathcal{F}},\ell_{\mathcal{G}}))/2}\|\nabla_x\ell_{\mathcal{G}}(x,y)\|_2\right)\right) \tag{25}$$
$$\leq \Pr\left(c'_{\mathcal{F}} \leq \epsilon\sqrt{(1+\overline{\mathcal{S}}(\ell_{\mathcal{F}},\ell_{\mathcal{G}}))/2}\|\nabla_x\ell_{\mathcal{F}}(x,y)\|_2\right) + \Pr\left(c'_{\mathcal{G}} \leq \epsilon\sqrt{(1+\overline{\mathcal{S}}(\ell_{\mathcal{F}},\ell_{\mathcal{G}}))/2}\|\nabla_x\ell_{\mathcal{G}}(x,y)\|_2\right), \tag{26}$$

where Eq. 23 comes from Eqs. 17 to 20, Eq. 24 comes from Eq. 21 and Eq. 22. The Eq. 25 is a result of Lemma 8: either

$$\delta \cdot \frac{\nabla_x \ell_{\mathcal{F}}(x, y)}{\|\nabla_x \ell_{\mathcal{F}}(x, y)\|_2} \leq \|\delta\|_2 \sqrt{(1 + \overline{\mathcal{S}}(\ell_{\mathcal{F}}, \ell_{\mathcal{G}}))/2}$$

or

$$\delta \cdot \frac{\nabla_x \ell_{\mathcal{G}}(x, y)}{\|\nabla_x \ell_{\mathcal{G}}(x, y)\|} \leq \|\delta\|_2 \sqrt{(1 + \overline{\mathcal{S}}(\ell_{\mathcal{F}}, \ell_{\mathcal{G}}))/2}.$$

We observe that by $\beta$-smoothness condition of the loss function,

$$c'_{\mathcal{F}} = \ell_{\mathcal{F}}(x^{\mathcal{A}}, y_t) - \ell_{\mathcal{F}}(x, y) - \frac{\beta}{2}\|\delta\|_2^2$$

$$\geq \ell_{\mathcal{F}}(x, y_t) + \delta \cdot \nabla_x \ell_{\mathcal{F}}(x, y_t) - \frac{\beta}{2}\|\delta\|_2^2 - \ell_{\mathcal{F}}(x, y) - \frac{\beta}{2}\|\delta\|_2^2.$$

Thus,

$$\Pr\left(c'_{\mathcal{F}} \leq \epsilon\sqrt{(1 + \overline{\mathcal{S}}(\ell_{\mathcal{F}}, \ell_{\mathcal{G}}))/2}\|\nabla_x \ell_{\mathcal{F}}(x, y)\|_2\right)$$

$$\leq \Pr\left(\ell_{\mathcal{F}}(x, y_t) - \ell_{\mathcal{F}}(x, y) \leq \epsilon B(1 + \sqrt{(1 + \overline{\mathcal{S}}(\ell_{\mathcal{F}}, \ell_{\mathcal{G}}))/2}) + \beta\epsilon^2\right)$$

$$\leq \Pr\left(\ell_{\mathcal{F}}(x, y) \geq \ell_{\mathcal{F}}(x, y_t) - \epsilon B(1 + \sqrt{(1 + \overline{\mathcal{S}}(\ell_{\mathcal{F}}, \ell_{\mathcal{G}}))/2} - \beta\epsilon^2\right) \qquad (27)$$

$$\leq \frac{\xi_{\mathcal{F}}}{\min\limits_{x \in \mathcal{X}} \ell_{\mathcal{F}}(x, y_t) - \epsilon B\left(1 + \sqrt{(1 + \overline{\mathcal{S}}(\ell_{\mathcal{F}}, \ell_{\mathcal{G}}))/2}\right) - \beta\epsilon^2}.$$

Similarly for $\mathcal{G}$,

$$\Pr\left(c'_{\mathcal{G}} \leq \epsilon\sqrt{(1 + \overline{\mathcal{S}}(\ell_{\mathcal{F}}, \ell_{\mathcal{G}}))/2}\|\nabla_x \ell_{\mathcal{G}}(x, y)\|_2\right)$$

$$\leq \frac{\xi_{\mathcal{G}}}{\min\limits_{x \in \mathcal{X}} \ell_{\mathcal{G}}(x, y_t) - \epsilon B\left(1 + \sqrt{(1 + \overline{\mathcal{S}}(\ell_{\mathcal{F}}, \ell_{\mathcal{G}}))/2}\right) - \beta\epsilon^2}. \qquad (28)$$

We conclude the proof by combining the above two equations into Eq. 26. $\qquad\square$

**Theorem** (Upper Bound on Untargeted Attack Transferability). *Assume both model $\mathcal{F}$ and $\mathcal{G}$ are $\beta$-smooth with gradient magnitude bounded by B, i.e., $\|\nabla_x \ell_{\mathcal{F}}(x, y)\| \leq B$ and $\|\nabla_x \ell_{\mathcal{G}}(x, y)\| \leq B$ for any $x \in \mathcal{X}, y \in \mathcal{Y}$. Let $\mathcal{A}_U$ be an $(\alpha, \mathcal{F})$-effective untargeted attack with perturbation ball $\|\delta\|_2 \leq \epsilon$. When the attack radius $\epsilon$ is small such that $\ell_{\min} - \epsilon B\left(1 + \sqrt{\frac{1 + \overline{\mathcal{S}}(\ell_{\mathcal{F}}, \ell_{\mathcal{G}})}{2}}\right) - \beta\epsilon^2 > 0$, the transferability can be upper bounded by*

$$\Pr\left(T_r(\mathcal{F}, \mathcal{G}, x) = 1\right) \leq \frac{\xi_{\mathcal{F}} + \xi_{\mathcal{G}}}{\ell_{\min} - \epsilon B\left(1 + \sqrt{\frac{1 + \overline{\mathcal{S}}(\ell_{\mathcal{F}}, \ell_{\mathcal{G}})}{2}}\right) - \beta\epsilon^2},$$

*where $\ell_{\min} = \min\limits_{\substack{x \in \mathcal{X}, y' \in \mathcal{Y}: \\ (x, y) \in \mathrm{supp}(\mathcal{D}), y' \neq y}} (\ell_{\mathcal{F}}(x, y'), \ell_{\mathcal{G}}(x, y'))$. Here $\xi_{\mathcal{F}}$ and $\xi_{\mathcal{G}}$ are the empirical risks of models $\mathcal{F}$ and $\mathcal{G}$ respectively, defined relative to a differentiable loss. The $\mathrm{supp}(\mathcal{D})$ is the support of benign data distribution, i.e., $x$ is the benign data and $y$ is its associated true label.*

*Proof.* The proof follows the proof for the targeted attack case. Accordingly, Eq. 21 and Eq. 22 are modified to

$$\delta \cdot \nabla_x \ell_{\mathcal{F}}(x, y) \geq \ell_{\mathcal{F}}(x^{\mathcal{A}}, y) - \ell_{\mathcal{F}}(x, y) - \frac{\beta}{2}\|\delta\|^2$$

$$\geq \ell_{\mathcal{F}}(x^{\mathcal{A}}, y_a) - \ell_{\mathcal{F}}(x, y) - \frac{\beta}{2}\|\delta\|^2 =: c'_{\mathcal{F}}. \qquad (29)$$

$$\delta \cdot \nabla_x \ell_{\mathcal{G}}(x, y) \geq \ell_{\mathcal{G}}(x^{\mathcal{A}}, y_b) - \ell_{\mathcal{G}}(x, y) - \frac{\beta}{2}\|\delta\|^2 =: c'_{\mathcal{G}} \qquad (30)$$

where $y_a$ and $y_b$ are the predicted labels of model $\mathcal{F}$ and $\mathcal{G}$ for $x^{\mathcal{A}}$ under a transferable untargeted attack respectively. Both $y_a$ and $y_b$ are not equal to $y$. Then, instead of $\min_{x \in \mathcal{X}} \ell_{\mathcal{F}/\mathcal{G}}(x, y_t)$ we use

$$\min_{x \in \mathcal{X}, y' \in \mathcal{Y}:(x,y) \in \mathrm{supp}(\mathcal{D}), y' \neq y} \ell_{\mathcal{F}/\mathcal{G}}(x, y')$$

in Eq. 27 and Eq. 28 and henceforth. $\qquad\square$

# E    Additional Details for Baseline Ensembles and Whitebox Attacks

In this section, we present a detailed introduction for our baseline ensembles and evaluated whitebox attacks. We introduce the baseline ensemble methods as follows:

- **Boosting** [35, 43] is a natural way of model ensemble training, which builds different weak learners in a sequential manner improving diversity in handling different task partitions. Here we consider two variants of boosting algorithms: 1) **AdaBoost** [19], where the final prediction will be the weighted average of all the weak learners: weight $\alpha_i$ for $i$-th base model is decided by the accumulated error $e_i$ as $\alpha_i = \log \frac{1-e_i}{e_i} + \log(K-1)$. Here $K$ refers to the number of categories in a classification task. As we can see, higher weight will be placed on stronger learners. 2) **GradientBoost** [16], which is a general ensemble training method by identifying weaker learners based on gradient information and generating the ensemble by training base models step by step with diverse learning orientations within pseudo-residuals $r = -\frac{\partial \ell(\bar{f}(x),y)}{\partial \bar{f}(x)}$ computed from the current ensemble model $f$ on input $x$ with ground truth label $y$.

- **CKAE** [25] develops diverse ensembles based on CKA measurement, which is recently shown to be effective to measure the orthogonality between representations. For two representations $K$ and $L$, $\mathrm{CKA}(K, L) = \frac{\mathrm{HSIC}(K,L)}{\sqrt{\mathrm{HSIC}(K,K)\mathrm{HSIC}(L,L)}}$, $\mathrm{HSIC}(K, L) = \frac{1}{(n-1)^2}\mathrm{tr}(KHLH)$, where $n$ is the number of samples and $H$ the centering matrix. For an ensemble consisting of base models $\{\mathcal{F}_i\}$, we regard the representation of $\mathcal{F}_i$ as its loss gradient vectors on batch samples and then minimize pair-wise CKA between base models $\mathcal{F}_i, \mathcal{F}_j$'s representations to encourage ensemble diversity.

- **ADP** [38] is proposed recently as an effective regularization-based training method to reduce adversarial transferability among base models within an ensemble by maximizing the volume spanned by base models' non-maximal output vectors. Specifically, for a ensemble consisting of base models $\{\mathcal{F}_i\}_{i=1}^N$ and input $x$ with ground truth label $y$, the ADP regularizer is defined as $\mathcal{L}_{\mathrm{ADP}}(x, y) = \alpha \cdot H(\mathrm{mean}(\{\mathcal{F}_i(x)\}_{i=1}^N)) + \beta \cdot \log(\mathbb{ED})$, where $H(\cdot)$ is the Shannon Entropy Loss and $\mathbb{ED}$ the square of the spanned volume. Nevertheless, the ADP ensemble has been shown to be vulnerable against attacks that run for enough iterations until converged [49]. We will also discuss this similar observation in our empirical robustness evaluation.

- **GAL** [23] promotes the diverse properties of the ensemble model by only minimizing the actual cosine similarities between pair-wise base models' loss gradient vectors. For $N$ base models $\{\mathcal{F}_i\}_{i=1}^N$ within an ensemble and input $x$ with ground truth label $y$, the GAL regularizer is defined as: $\mathcal{L}_{\mathrm{GAL}} = \log(\sum_{1 \leq i < j \leq N} \exp(CS(\nabla_x \ell_{\mathcal{F}_i}, \nabla_x \ell_{\mathcal{F}_j}))$ where $CS(\cdot, \cdot)$ refers to the actual cosine similarity measurement and $\nabla_x \ell_{\mathcal{F}_i}$ the loss gradient of base model $\mathcal{F}_i$ on $x$. It could serve as a baseline to empirically verify our theoretical analysis: when the loss gradients of base models are similar, the smoother the base models are, the less transferable they are.

- **DVERGE** [56] reduces the transferability among base models by utilizing Cross-Adversarial-Training: For a ensemble consisting of base models $\{\mathcal{F}_i\}$ and input $x$ with ground truth label $y$, each base model $\mathcal{F}_i$ is trained with the non-robust feature instances [22] generated against another base model. Specifically, DVERGE minimizes $\sum_{j \neq i} \ell(\mathcal{F}_i(x'_{\mathcal{F}_j}(x_s, x)), y_s)$ for every $\mathcal{F}_i$ iteratively, where $x'_{\mathcal{F}_j}(x_s, x)$ represents the non-robust features against $\mathcal{F}_j$ based on the randomly chosen input $(x_s, y_s)$. $\ell(\cdot, \cdot)$ is the cross-entropy loss function.

We consider the following attacks for whitebox robustness evaluation. Here we define $(x, y)$ to be the input $x$ with label $y$ and $x^{\mathcal{A}}$ to be the notion of adversarial example generated from $x$. $\ell(\mathcal{F}(x), y)$ refers to the loss between model output $\mathcal{F}(x)$ and label $y$, and $\epsilon$ is the $\ell_\infty$ perturbation magnitude bound for different attacks.

- *Fast Gradient Sign Method* (FGSM) [17] is a simple yet effective attack strategy which generates adversarial example $x^{\mathcal{A}} = x + \nu$ by assigning $\nu = \epsilon \cdot \mathrm{sgn}(\nabla_x \ell(\mathcal{F}(x), y))$.

- *Basic Iterative Method* (BIM) [34] is an iterative attack method which adds adversarial perturbations step by step: $x_{i+1} = \text{clip}(x_i + \alpha \cdot \nabla_{x_i}\ell(\mathcal{F}(x_i), y))$, with initial starting point $x_0 = x$. Function $\text{clip}(\cdot)$ projects the perturbed instance back to the $\ell_\infty$ ball within the perturbation range $\epsilon$, and $\alpha$ refers to the step size.

- *Momentum Iterative Method* (MIM) [13] can be regarded as the variant of BIM by utilizing the gradient momentum during the iterative attack procedure. Within iteration $i + 1$, we update new gradient as $g_{i+1} = \mu g_i + \frac{\ell(\mathcal{F}(x_i), y)}{\|\nabla_x \ell(\mathcal{F}(x_i), y)\|_1}$ and set $x_{i+1} = \text{clip}(x_i + \alpha \cdot g_{i+1})$ while $\mu$ refers to the momentum coefficient and $\alpha$ the step size.

- *Projected Gradient Descent* (PGD) [34] can be regarded as the variant of BIM by sampling $x_0$ randomly within the $\ell_p$ ball around $x$ within radius $\epsilon$. After initialization, it follows the standard BIM procedure by setting $x_{i+1} = \text{clip}(x_i + \alpha \cdot \nabla_{x_i}\ell(\mathcal{F}(x_i), y))$ on $i$-th attack iteration.

- *Auto-PGD* (APGD) [11] is a step-size free variant of PGD by configuring the step-size according to the overall iteration budgets and the progress of the current attack. Here we consider APGD-CE and APGD-DLR attack which use CrossEntropy (CE) and Difference of Logits Ratio (DLR) [11] loss as their loss function correspondingly.

- *Carlini & Wanger Attack* (CW) [5] accomplishes the attack by solving the optimization problem: $x^{\mathcal{A}} := \min_{x'} \|x' - x\|_2^2 + c \cdot f(x', y)$, where $c$ is a constant to balance the perturbation scale and attack success rate, and $f$ is the adversarial attack loss designed to satisfy the sufficient and necessary condition of different attacks. For instance, the untargeted attack loss is represented as $f(x', y) = \max(\mathcal{F}(x)_y - \mathcal{F}(x)_{i \neq y}, -\kappa)$ while $\kappa$ is a confidence variable with value $0.1$ as default.

- *Elastic-net Attack* (EAD) [8] follows the similar optimization of CW Attack while considering both $\ell_2$ and $\ell_1$ distortion: $x^{\mathcal{A}} := \min_{x'} \|x' - x\|_2^2 + \beta\|x' - x\|_1 + c \cdot f(x', y)$. Here $\beta, c$ refer to the balancing parameters and $f(x', y) = \max(\mathcal{F}(x)_y - \mathcal{F}(x)_{i \neq y}, -\kappa)$ under untargeted attack setting. We set $\beta = 0.01, \kappa = 0.1$ as default.

In our experiments, we set 50 attack iterations with step size $\alpha = \epsilon/5$ for BIM, MIM attack and PGD attack with 5 random starts. For CW and EAD attacks, we set the number of attack iterations as 1000 and evaluate them with different constant $c$ for different datasets.

## F  Training Details

We adapt ResNet-20 [20] as the base model architecture and Adam optimizer [24] in all of our experiments.

**TRS training algorithm.** We show the one-epoch TRS training algorithm pseudo code in Algorithm 1. We apply the mini-batch training strategy and train the TRS ensemble for $M$ epochs ($M = 120$ for MNIST and $M = 200$ for CIFAR-10) in our experiments. To decide the $\delta$ within the local min-max procedure, we use the **Warm-up** strategy by linearly increasing the local $\ell_\infty$ ball's radius $\delta$ from small initial $\delta_0$ to the final $\delta_M$ along with the increasing of training epochs.

**Baseline training details.** For ADP and GAL, we follow the exact training configuration mentioned in their paper in both MNIST and CIFAR-10 experiments. For DVERGE, we set the same feature distillation $\epsilon = 0.07$ with step size as $0.007$ for CIFAR-10 as they mentioned in their paper but set $\epsilon = 0.5$ with step size as $0.05$ for MNIST since they did not conduct any MNIST experiments in their paper. We set training epochs as 120 for MNIST and 200 for CIFAR-10 and CIFAR-100 in baseline training.

**TRS training details.** For MNIST, we set the initial learning rate $\alpha = 0.001$ and train our TRS ensemble for 120 epochs by decaying the learning rate by $0.1$ at 40-th and 80-th epochs. For CIFAR-10 and CIFAR-100 we set the initial learning rate $\alpha = 0.001$ and train our TRS ensemble for 200 epochs by decaying the learning rate by $0.1$ at 100-th and 150-th epochs. For PGD Optimization within $\mathcal{L}_{\text{smooth}}$ approximation, we set step size $\tilde{\alpha} = \delta/3$ and the total number of steps $T$ as 6 for both MNIST and CIFAR-10 experiments. We also leverage the ablation study about the convergence of PGD optimization w.r.t the robustness of TRS ensemble by varying $\tilde{\alpha}$ and $T$ in I.5.

By configuring the default TRS ensemble training setting, we evaluated the average epoch training time for TRS and compared it to other baselines (ADP, GAL, DVERGE) on RTX 2080 single GPU device. Results are shown in Table 2.

**Algorithm 1** TRS training framework in epoch $m$ for an ensemble with $N$ base models $\{\mathcal{F}_i\}$, with the total number of training epochs $M$.

1: $\delta_m \leftarrow \delta_0 + (\delta_M - \delta_0) \cdot m/M$
2: **for** $b = 1, \cdots, B$ **do**
3: $\quad (x, y) \leftarrow$ training instances from $b$-th mini-batch
4: $\quad \mathcal{L}_{\text{Reg}} \leftarrow 0$
5: $\quad \mathcal{L}_{\text{ECE}} \leftarrow 0$
6: $\quad$ **for** $i = 1, \cdots, N$ **do**
7: $\quad\quad$ **for** $j = i + 1, \cdots, N$ **do**
8: $\quad\quad\quad \mathcal{L}_{\text{Reg}} \leftarrow \mathcal{L}_{\text{Reg}} + \mathcal{L}_{\text{TRS}}(\mathcal{F}_i, \mathcal{F}_j, x, \delta_m)$
9: $\quad\quad$ **end for**
10: $\quad$ **end for**
11: $\quad$ **for** $i = 1, \cdots, N$ **do**
12: $\quad\quad \mathcal{L}_{\text{ECE}} \leftarrow \mathcal{L}_{\text{ECE}} + \mathcal{L}_{\text{CE}}(\mathcal{F}_i(x), y)$
13: $\quad$ **end for**
14: $\quad \mathcal{L}_{\text{Reg}} \leftarrow \mathcal{L}_{\text{Reg}}/\binom{N}{2}$
15: $\quad \mathcal{L}_{\text{ECE}} \leftarrow \mathcal{L}_{\text{ECE}}/N$
16: $\quad$ **for** $i = 1, \cdots, N$ **do**
17: $\quad\quad \nabla_{\mathcal{F}_i} \leftarrow \nabla_{\mathcal{F}_i}[\mathcal{L}_{\text{ECE}} + \mathcal{L}_{\text{Reg}}]$
18: $\quad\quad \mathcal{F}_i \leftarrow \mathcal{F}_i - lr \cdot \nabla_{\mathcal{F}_i}$
19: $\quad$ **end for**
20: **end for**

Table 2: Comparison on average epoch training time (s) between TRS training and other baseline training methods, evaluated on RTX 2080 single GPU device.

| Avg epoch training time (s) | ADP | GAL | DVERGE | TRS |
|---|---|---|---|---|
| MNIST | 29.22 | 106.81 | 184.42 | 302.24 |
| CIFAR-10 | 33.22 | 139.10 | 349.61 | 1291.55 |

Our results show that though ADP, GAL require less training time, they can not achieve even comparable robustness with TRS as shown in our paper. Compared with DVERGE, TRS requires longer training time but maintains higher robustness under almost all attack scenarios.

## G   Numerical Results of Blackbox Robustness Evaluation

Table 3 and 4 show the detailed robust accuracy number of different ensembles against blackbox transfer attack with different perturbation scale $\epsilon$, which corresponds to the Figure 2. As we can see, TRS ensemble shows its competitive robustness to DVERGE on small $\epsilon$ setting but much better stability of robustness on large $\epsilon$ setting although it slightly sacrifices benign accuracy on clean data.

## H   Statistical Stability Analysis on Robust Accuracy

For attacks with random-start (PGD, APGD-DLR, APGD-CE) mentioned in Table 1, we run each of them 10 times with different random seeds and evaluate them on TRS ensemble to present the statistical indicators (Min, Max, Mean, Std) of robust accuracy in Table 6. We can conclude that our reported robust accuracy shows statistical stability given the standard deviation is smaller than 0.3 under all the scenarios.

## I   Ablation Studies

### I.1   Decision Boundary Analysis

We visualize the decision boundary of the GAL, DVERGE and TRS ensembles for MNIST and CIFAR-10 in Figure 4. The dashed line is the negative gradient direction and the horizontal direction

Table 3: Robust accuracy (%) of different approaches against **blackbox transfer attack** with different perturbation scales $\epsilon$ on MNIST dataset.

| $\epsilon$ | clean | 0.10 | 0.15 | 0.20 | 0.25 | 0.30 | 0.35 | 0.40 |
|---|---|---|---|---|---|---|---|---|
| Vanilla | 99.5 | 1.8 | 0.1 | 0.0 | 0.0 | 0.0 | 0.0 | 0.0 |
| ADP | 99.4 | 25.5 | 13.8 | 7.0 | 2.1 | 0.3 | 0.1 | 0.0 |
| GAL | 98.7 | 96.8 | 77.0 | 29.1 | 12.8 | 4.6 | 1.9 | 0.6 |
| DVERGE | 98.7 | **97.6** | **97.4** | **96.9** | 96.2 | 94.2 | 78.3 | 20.2 |
| TRS | 98.6 | 97.2 | 96.7 | 96.5 | **96.3** | **95.5** | **93.1** | **86.4** |

Table 4: Robust accuracy (%) of different approaches against **blackbox transfer attack** with different perturbation scales $\epsilon$ on CIFAR-10 dataset.

| $\epsilon$ | clean | 0.01 | 0.02 | 0.03 | 0.04 | 0.05 | 0.06 | 0.07 |
|---|---|---|---|---|---|---|---|---|
| Vanilla | 94.1 | 10.0 | 0.1 | 0.0 | 0.0 | 0.0 | 0.0 | 0.0 |
| ADP | 91.6 | 20.7 | 0.5 | 0.0 | 0.0 | 0.0 | 0.0 | 0.0 |
| GAL | 88.3 | 74.6 | 58.9 | 39.1 | 22.0 | 11.3 | 5.2 | 2.1 |
| DVERGE | 91.9 | **83.3** | 69.0 | 49.8 | 28.2 | 14.4 | 4.0 | 0.8 |
| TRS | 86.7 | 82.3 | **76.1** | **65.8** | **55.0** | **45.5** | **35.8** | **26.7** |

Table 5: Robust accuracy (%) of TRS ensemble trained with different hyper-parameter settings against various whitebox attacks on MNIST dataset.

| $\lambda_a$ | | 100 | | | | 500 | | | |
|---|---|---|---|---|---|---|---|---|---|
| $\lambda_b$ | | 2.5 | | 10 | | 2.5 | | 10 | |
| $\delta_M$ | | 0.3 | 0.4 | 0.3 | 0.4 | 0.3 | 0.4 | 0.3 | 0.4 |
| FGSM | $\epsilon = 0.1$ | **95.6** | 94.6 | 90.6 | 94.8 | **95.6** | 93.0 | 95.2 | 93.6 |
| | $\epsilon = 0.2$ | 91.7 | 83.4 | 89.7 | 87.3 | **92.0** | 84.0 | 88.0 | 85.0 |
| BIM (50) | $\epsilon = 0.1$ | **93.3** | 82.9 | 75.7 | 92.5 | 88.2 | 83.9 | 92.6 | 90.9 |
| | $\epsilon = 0.15$ | **85.7** | 69.7 | 61.3 | 84.1 | 73.1 | 61.3 | 82.2 | 83.3 |
| PGD (50) | $\epsilon = 0.1$ | **93.0** | 79.1 | 74.3 | 92.2 | 86.3 | 83.3 | 91.7 | 90.6 |
| | $\epsilon = 0.15$ | **85.1** | 62.6 | 57.4 | 82.6 | 69.9 | 58.2 | 80.0 | 82.9 |
| MIM (50) | $\epsilon = 0.1$ | **92.9** | 81.6 | 75.1 | 92.0 | 87.7 | 83.5 | 91.7 | 91.2 |
| | $\epsilon = 0.15$ | **85.1** | 68.2 | 60.2 | 83.7 | 74.0 | 62.4 | 82.4 | 83.4 |
| CW | $c = 0.1$ | 98.1 | 96.6 | 96.4 | 97.5 | **98.4** | 97.2 | 98.1 | 97.8 |
| | $c = 1.0$ | 92.6 | 92.6 | 89.1 | **95.9** | 86.1 | 77.4 | 88.2 | 95.1 |
| EAD | $c = 1.0$ | 23.3 | 14.3 | 9.2 | **24.1** | 22.5 | 2.6 | 3.4 | 23.9 |
| | $c = 5.0$ | 1.4 | 0.9 | 0.1 | **2.3** | 0.0 | 0.0 | 0.2 | 1.7 |
| APGD-DLR | $\epsilon = 0.1$ | **92.1** | 78.5 | 72.8 | 91.5 | 85.9 | 82.8 | 91.1 | 90.2 |
| | $\epsilon = 0.15$ | **83.4** | 62.1 | 57.0 | 82.3 | 69.6 | 57.9 | 79.8 | 82.4 |
| APGD-CE | $\epsilon = 0.1$ | **91.7** | 78.1 | 72.1 | 91.2 | 85.2 | 82.5 | 90.8 | 89.7 |
| | $\epsilon = 0.15$ | **82.8** | 61.3 | 56.5 | 81.9 | 69.3 | 57.6 | 79.4 | 81.7 |

Table 6: {Min, Max, Mean, Std} of Robust accuracy (%) of TRS ensemble against 10 times whitebox attacks simulation with different random seeds on MNIST and CIFAR-10 datasets.

| Robust Accuracy | | param. | Min | Max | Mean | Std |
|---|---|---|---|---|---|---|
| MNIST | PGD | $\epsilon = 0.1$ | 92.8 | 93.2 | 93.1 | 0.143 |
| | | $\epsilon = 0.15$ | 84.9 | 85.1 | 85.1 | 0.067 |
| | APGD-DLR | $\epsilon = 0.1$ | 92.1 | 92.3 | 92.2 | 0.083 |
| | | $\epsilon = 0.15$ | 83.2 | 83.5 | 83.4 | 0.114 |
| | APGD-CE | $\epsilon = 0.1$ | 91.7 | 92.0 | 91.9 | 0.102 |
| | | $\epsilon = 0.15$ | 82.5 | 82.9 | 82.7 | 0.120 |
| CIFAR-10 | PGD | $\epsilon = 0.01$ | 50.4 | 50.5 | 50.4 | 0.049 |
| | | $\epsilon = 0.02$ | 14.8 | 15.8 | 15.2 | 0.293 |
| | APGD-DLR | $\epsilon = 0.01$ | 50.0 | 50.5 | 50.2 | 0.151 |
| | | $\epsilon = 0.02$ | 15.2 | 16.0 | 15.6 | 0.234 |
| | APGD-CE | $\epsilon = 0.01$ | 48.6 | 48.9 | 48.8 | 0.090 |
| | | $\epsilon = 0.02$ | 15.3 | 16.0 | 15.6 | 0.199 |

is randomly chosen which is orthogonal to the gradient direction. From the decision boundary of GAL ensemble, we can see that controlling only the gradient similarity will lead to a very non-smooth

Table 7: Conditional Robust Accuracy (%) of Adversarial Training based ensemble (AdvT) and TRS+AdvT ensemble against (Top) **whitebox attacks** and (Down) **blackbox attack** with different perturbation scales $\epsilon$.

| Attacks | | FGSM | | BIM (50) | | PGD (50) | | MIM (50) | |
|---|---|---|---|---|---|---|---|---|---|
| MNIST | $\epsilon$ | 0.10 | 0.20 | 0.10 | 0.15 | 0.10 | 0.15 | 0.10 | 0.15 |
| | AdvT | 98.4 | 97.3 | 98.2 | 97.5 | 98.2 | 97.2 | 98.2 | 97.6 |
| | TRS+AdvT | **99.1** | **98.0** | **99.0** | **98.2** | **98.9** | **98.0** | **99.0** | **98.1** |
| CIFAR-10 | $\epsilon$ | 0.02 | 0.04 | 0.01 | 0.02 | 0.01 | 0.02 | 0.01 | 0.02 |
| | AdvT | **79.3** | **60.0** | 88.5 | 76.1 | 88.4 | 76.1 | 88.5 | 76.3 |
| | TRS+AdvT | 79.2 | 58.0 | **90.7** | **76.7** | **90.7** | **76.6** | **90.9** | **76.9** |

| | $\epsilon$ | 0.10 | 0.15 | 0.20 | 0.25 | 0.30 | 0.35 | 0.40 |
|---|---|---|---|---|---|---|---|---|
| MNIST | AdvT | 98.9 | 98.7 | 98.6 | 98.4 | 98.4 | 91.6 | 8.1 |
| | TRS+AdvT | **99.4** | **99.3** | **99.1** | **99.1** | **98.9** | **98.7** | **98.5** |
| CIFAR-10 | $\epsilon$ | 0.01 | 0.02 | 0.03 | 0.04 | 0.05 | 0.06 | 0.07 |
| | AdvT | 98.4 | 96.2 | 93.9 | 91.5 | 89.0 | 84.9 | 81.1 |
| | TRS+AdvT | **98.8** | **97.7** | **94.9** | **92.6** | **89.9** | **86.3** | **81.6** |

model decision boundary and thus harm the model robustness. From the comparison of DVERGE and TRS ensemble, we find that DVERGE ensemble tends to be more robust along the gradient direction especially on CIFAR-10, (i.e. the distance to the boundary is larger and sometimes even larger than along the other random direction). This may be due to the reason that DVERGE is essentially performing adversarial training for different base models and therefore it protects the adversarial (gradient) direction. Thus, DVERGE performs better against weak attacks which only consider the gradient direction (e.g. FGSM on CIFAR-10). On the other hand, we find that TRS training yields a smoother model along different directions than DVERGE, which leads to more consistent predictions within a large neighborhood of an input. Thus, the TRS ensemble has higher robustness in different directions against strong attacks such as PGD attack.

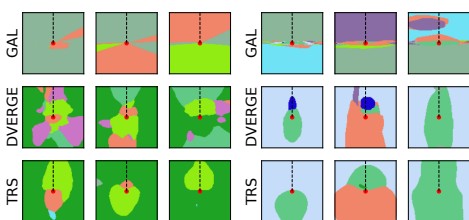

Figure 4: The decision boundary of different models around testing images on (left) MNIST and (right) CIFAR-10 dataset. Same color indicates the same model prediction. The dash lines shows the negative gradient direction, which is used in the gradient-based attacks.

## I.2    TRS with Adversarial Training

While the TRS regularizer could reduce adversarial transferability among base models by enforcing low similarity on loss gradients and promoting model smoothness, we explore whether Adversarial Training [34], which aims to reduce base models' vulnerability, is able to further improve the robustness of TRS or not. We first apply adversarial training to train an ensemble model (AdvT) containing 3 base models as in TRS ensemble. During training, we use $\ell_\infty$ adversarial perturbation $\delta_{\text{adv}}$ ($\|\delta_{\text{adv}}\|_\infty \leq 0.2$ for MNIST and $\|\delta_{\text{adv}}\|_\infty \leq 0.03$ for CIFAR-10). To combine TRS with adversarial training (TRS+AdvT), we combine TRS regularizer Loss $\mathcal{L}_{\text{TRS}}$ with Adversarial Training Loss $\mathcal{L}_{\text{AdvT}} = \max_{\|x-x'\|_\infty \leq \delta_{\text{adv}}} \ell_{\mathcal{F}}(x', y)$ on input $x$ with label $y$ with the same weight, and train the ensemble $\mathcal{F}$ jointly. We evaluate both whitebox and blackbox robustness of TRS+AdvT and AdvT ensembles. During the evaluation, we consider the *Conditional Robust Accuracy* evaluated on adversarial examples generated based on correctly classified clean samples to eliminate the influence of model benign accuracy. Other settings are the same as we have introduced in Section 4.1.

Table 7 shows the robustness of both AdvT and TRS+AdvT under whitebox and blackbox attacks on different datasets. As we can see, TRS+AdvT ensemble outperforms the traditional adversarial training based ensemble consistently especially when $\epsilon$ is large.

## I.3  Impacts of $\mathcal{L}_{\text{sim}}$ and $\mathcal{L}_{\text{smooth}}$

To better understand the exact effects of regularizing $\mathcal{L}_{\text{sim}}$ and $\mathcal{L}_{\text{smooth}}$, we conduct ablation studies by regularizing $\mathcal{L}_{\text{sim}}$ or $\mathcal{L}_{\text{smooth}}$ only on both MNIST and CIFAR-10 datasets. Results are shown in Table 8.

We can see that, though training with $\mathcal{L}_{\text{smooth}}$ only could lead to high robustness, TRS ensemble could achieve even higher robustness against strong multi-step attacks by concerning similarity loss $\mathcal{L}_{\text{sim}}$ at the same time. This indicates that both model smoothness and model diversity are important, though $\mathcal{L}_{\text{smooth}}$ would take the majority.

Table 8: Robust accuracy (%) of TRS ensemble trained by regularizing $\mathcal{L}_{\text{sim}}$ or $\mathcal{L}_{\text{smooth}}$ only, or together, against various white-box attacks on MNIST and CIFAR-10 datasets.

| Robust Accuracy | | FGSM | BIM | PGD | MIM | CW | EAD |
|---|---|---|---|---|---|---|---|
| | param. | $\epsilon = 0.2$ | $\epsilon = 0.15$ | $\epsilon = 0.15$ | $\epsilon = 0.15$ | $c = 1.0$ | $c = 10.0$ |
| MNIST | $\mathcal{L}_{\text{sim}}$ only | 30.7 | 0.0 | 0.0 | 0.0 | 58.6 | 0.5 |
| | $\mathcal{L}_{\text{smooth}}$ only | **93.1** | 82.5 | 80.7 | 82.6 | 86.2 | 1.2 |
| | $\mathcal{L}_{\text{sim}} + \mathcal{L}_{\text{smooth}}$ (TRS) | 91.7 | **85.7** | **85.1** | **85.1** | **92.6** | **1.4** |
| | param. | $\epsilon = 0.04$ | $\epsilon = 0.02$ | $\epsilon = 0.02$ | $\epsilon = 0.02$ | $c = 1.0$ | $c = 5.0$ |
| CIFAR-10 | $\mathcal{L}_{\text{sim}}$ only | **35.0** | 0.0 | 0.0 | 0.0 | 17.6 | 0.0 |
| | $\mathcal{L}_{\text{smooth}}$ only | 9.3 | 13.9 | 13.8 | 15.0 | 43.0 | 0.0 |
| | $\mathcal{L}_{\text{sim}} + \mathcal{L}_{\text{smooth}}$ (TRS) | 24.9 | **15.8** | **15.1** | **17.2** | **58.1** | **0.1** |

## I.4  Robust Accuracy Convergence Analysis

We observe that when the number of attack iterations is large, both ADP and GAL regularizer trained ensembles achieve much lower robust accuracy against iterative attacks (BIM, PGD, MIM) than the reported robustness in the original papers which is estimated under a small number of attack iterations. This case implies the non-convergence of iterative attack evaluation mentioned in their papers, which is also confirmed by [49]. In contrast, both DVERGE and TRS still remain highly robust against iterative attacks with large iterations. To show the stability of our model's robust accuracy, we evaluate it against PGD attack with 500 and 1000 attack iterations. Results are shown in Table 9 where TRS ensemble's robust accuracy only slightly drops after increasing the attack iterations, and outperforms DVERGE by a large margin.

## I.5  Convergence of PGD Optimization within $\mathcal{L}_{\text{smooth}}$ Approximation

Since the computation cost of training a TRS ensemble partially relies on the complexity of PGD procedure on solving the inner-maximization task of $\mathcal{L}_{\text{smooth}}$, we conduct ablation study on analyzing the trade-off between the computation cost (by varying PGD steps $T$) and the resulting robustness of TRS ensemble on MNIST dataset. Specifically, we consider the following settings of PGD step size $\tilde{\alpha}$ and the number of steps $T$:

(1)  $\tilde{\alpha} = \delta, T = 1$

(2)  $\tilde{\alpha} = \delta/3, T = 6$

(3)  $\tilde{\alpha} = \delta/10, T = 20$

Results are shown in Table 10. As we can see, the robustness of TRS ensemble consistently improves with the increasing of $T$ and converges. We should also notice that, even for $T = 1$, TRS ensemble is more robust than the strongest baseline DVERGE against various strong attacks. Due to the positive correlation between $T$ and the training cost, we should choose suitable $T$ balancing the training cost and model robustness. For MNIST, the default setting ($T = 6$) could be a good choice.

# J  Robustness of TRS Ensemble against Other Strong Blackbox Attacks

We also conduct additional blackbox robustness evaluation against the following three strong blackbox attacks which focus on attack transferability between surrogate model and target model:

Table 9: Convergence of PGD attack on different ensembles.

| Settings | | iters | ADP | GAL | DVERGE | TRS |
|---|---|---|---|---|---|---|
| MNIST | $\epsilon = 0.10$ | 50 | 4.5 | 4.1 | 69.2 | **93.0** |
| | | 500 | 1.6 | 1.1 | 66.5 | **92.8** |
| | | 1000 | 1.6 | 1.0 | 66.3 | **92.6** |
| | $\epsilon = 0.15$ | 50 | 1.0 | 0.6 | 28.8 | **85.1** |
| | | 500 | 0.5 | 0.1 | 25.0 | **83.6** |
| | | 1000 | 0.4 | 0.1 | 24.8 | **83.5** |
| CIFAR-10 | $\epsilon = 0.01$ | 50 | 9.0 | 8.3 | 37.1 | **50.5** |
| | | 500 | 3.5 | 7.8 | 35.8 | **50.3** |
| | | 1000 | 2.9 | 7.8 | 35.7 | **50.2** |
| | $\epsilon = 0.02$ | 50 | 0.1 | 0.6 | 10.5 | **15.1** |
| | | 500 | 0.0 | 0.3 | 9.0 | **14.5** |
| | | 1000 | 0.0 | 0.3 | 8.8 | **14.5** |

Table 10: Robustness of TRS ensemble against various white-box attacks by varying PGD step size $\tilde{\alpha}$ and total number of steps $T$ for solving the inner-maximization within $\mathcal{L}_{\text{smooth}}$ on MNIST dataset.

| Robust acc on MNIST | FGSM | BIM | PGD | MIM | CW | EAD |
|---|---|---|---|---|---|---|
| param. | $\epsilon = 0.2$ | $\epsilon = 0.15$ | $\epsilon = 0.15$ | $\epsilon = 0.15$ | $c = 1.0$ | $c = 10.0$ |
| DVERGE | 91.6 | 47.7 | 28.8 | 44.6 | 79.2 | 0.0 |
| TRS ($\tilde{\alpha} = \delta, T = 1$) | 90.5 | 76.0 | 70.1 | 73.3 | 89.6 | 0.1 |
| TRS ($\tilde{\alpha} = \delta/3, T = 6$) | 91.7 | 85.7 | 85.1 | 85.1 | 92.6 | **1.4** |
| TRS ($\tilde{\alpha} = \delta/10, T = 20$) | **92.1** | **87.2** | **85.5** | **86.1** | 92.8 | **1.4** |

Table 11: Robust accuracy (%) of different approaches against strong blackbox transfer attack on MNIST and CIFAR-10 datasets.

| Robust Accuracy | | ILA | DI2-FGSM | IRA |
|---|---|---|---|---|
| MNIST ($\epsilon = 0.3$) | ADP | 5.4 | 9.9 | 4.8 |
| | GAL | 3.0 | 8.6 | 7.1 |
| | DVERGE | 89.5 | 91.6 | 82.0 |
| | TRS | **91.2** | **93.7** | **84.4** |
| CIFAR-10 ($\epsilon = 0.05$) | ADP | 1.2 | 1.6 | 1.4 |
| | GAL | 32.2 | 36.2 | 29.2 |
| | DVERGE | 35.9 | 38.3 | 32.4 |
| | TRS | **46.2** | **50.0** | **45.1** |

- *Intermediate Level Attack* (ILA) [21] enhances the blackbox attack transferability by taking the perturbation on an intermediate layer of surrogate model into account.

- *DI2-FGSM* [55] can be viewed as a variant of BIM by applying input transformation randomly at each attack iteration to promote diverse input patterns.

- *Interaction Reduced Attack* (IRA) [50] integrates an additional interaction loss term after analyzing the negative correlation between attack transferability and interaction between adversarial units.

We use the open-source code mentioned in their original papers and generate blackbox adversarial examples from a surrogate ensemble model consisting of three ResNet20 submodels for both MNIST and CIFAR-10 datasets. We compare the robustness of TRS ensemble with other baseline ensemble. Results are shown in Table 11.

We can find that, TRS ensemble consistently demonstrates the highest robustness compared to other baseline ensembles, which indicates solid blackbox robustness of TRS ensemble against various types of blackbox attacks.

# K  Robustness of TRS Ensemble on CIFAR-100 Dataset

Besides MNIST and CIFAR-10 datasets, we also evaluate our proposed TRS ensemble on the CIFAR-100 dataset. The base model structure and training parameter configuration remain the same as in CIFAR-10 experiments. The whitebox robustness evaluation results are shown in Table 12. From the results, we can see that the robustness of TRS model is better than other methods against all attacks except FGSM, which is similar with our observations in CIFAR-10. This shows that our TRS algorithm still achieves a good performance on classification tasks with large number of classes.

Table 12: Robust accuracy(%) of different ensembles against whitebox attacks on CIFAR-100. "para." refers to the attack parameter ($\epsilon$ is the $\ell_\infty$ perturbation budget for the attack and $c$ the constant to balance the attack stealthiness and effectiveness).

| CIFAR-100 | para. | ADP | GAL | DVERGE | TRS |
|---|---|---|---|---|---|
| FGSM | $\epsilon = 0.02$ | 11.5 | 28.7 | **29.7** | 19.3 |
| | $\epsilon = 0.04$ | 6.4 | 2.7 | **25.4** | 9.5 |
| BIM (50) | $\epsilon = 0.01$ | 0.5 | 7.6 | 12.1 | **22.9** |
| | $\epsilon = 0.02$ | 0.0 | 1.5 | 2.9 | **5.4** |
| PGD (50) | $\epsilon = 0.01$ | 0.4 | 5.4 | 11.3 | **23.0** |
| | $\epsilon = 0.02$ | 0.0 | 1.1 | 2.0 | **5.3** |
| MIM (50) | $\epsilon = 0.01$ | 0.5 | 5.7 | 13.1 | **23.4** |
| | $\epsilon = 0.02$ | 0.0 | 0.5 | 2.6 | **6.2** |
| CW | $c = 0.01$ | 11.3 | 32.0 | 44.8 | **45.7** |
| | $c = 0.1$ | 0.5 | 10.7 | 20.3 | **26.9** |
| EAD | $c = 1.0$ | 0.0 | 0.0 | 1.0 | **5.7** |
| | $c = 5.0$ | 0.0 | 0.0 | 0.0 | **0.3** |
| APGD-LR | $\epsilon = 0.01$ | 0.2 | 4.3 | 11.8 | **22.2** |
| | $\epsilon = 0.02$ | 0.0 | 0.6 | 2.1 | **5.3** |
| APGD-CE | $\epsilon = 0.01$ | 0.2 | 4.2 | 11.3 | **20.7** |
| | $\epsilon = 0.02$ | 0.0 | 0.4 | 1.7 | **4.8** |