# OpenReview forum: "TRS: Transferability Reduced Ensemble via Promoting Gradient Diversity and Model Smoothness"
_NeurIPS.cc/2021/Conference — NeurIPS 2021 Poster_

### Official Review · Reviewer_73Cj · 2021-07-12

**Rating:** 7
**Confidence:** 5

**Summary:**

1. This paper provides theoretical analysus on the sufficient condition on the adversarial transferability between ML models, and identify the importance of both gradient orthogonality and model smoothness

2. This paper proposes a simple smoothness regularization term and merge it with gradient diversity objective to form TRS robust ensemble training algorithm

3. Thorough experiments are provided to show TRS achieves state-of-the-art ensemble robustness

**Ethical Concerns:**

There's no ethical concern related to this submission

**Limitations And Societal Impact:**

One potential limitation of the proposed method is the computation cost of optimizing the min-max smoothness objective in Equ. (1). Given the large performance margin between using the min-max objective vs. using vanilla L2, it is likely that the perofrmance of TRS is determined by how well the inner maximization is optimized, yet the discussion on this part is lacking in the paper. I would suggest the author to
1. Provide detailed information on the hyperparameter (i.e. step size and number of steps for the inner maximization) of the smoothness loss and discuss its impact on the overall training cost
2. If possible provide an ablation study on how these hyperparameters affect the final performance of TRS, is there a tradeoff in the training cost vs. robustness?

**Main Review:**

This paper provides solid theortical analysis on the transfer robustness of deep learning models, and accordingly propose ensemble training methods to reduce the transferability between sub-models and achieve robust ensemble model. Overall I enjoy reading this paper and find it interesting. Here are the strength and weakness of the paper:

Strength:
1. The theortical analysis on the transferability bound is solid, and its the first work to identify the importance of model smoothness in diverse ensemble training, making the proposed TRS training method well motivated.
2. The experiment results support the theortical claims by showing the ensemble trained with TRS can achieve SOTA transferability robutness and ensemble robustness.
3. The paper is well written and easy to follow.

Weakness:
1. There lacks a discussion on the additional training cost introduced by optimizing the smoothness loss. See the limitation section for details.
2. It should be noted that model smoothness has already been introduced in previous works on improving the robustness of deep learning model. One representative work is CURE [1], which also use a form of gradient regularization to minimize local curvature of deep learning model for robusntess improvement. I think it would be important for the author to cite this paper and discuss the difference in the smoothness objective.
3.  Though I appreciate the finding of the importance of model smoothness , how exactly the smoothness regularization is contributing to the overall ensemble robustness is unclear. The author suggest the smoothness is helpful on achieving better model diversity. However, as shown in CURE, adding smoothness regularization is also an effective way to increase the robustness of individual model. So it's likely that TRS is achieving higher robustness against larger perturbation due to each sub-mode is more robust. This hypothesis could also be supported by the fact that the the curve of TRS in Fig 2 looks more like DVERGE+ADVT (lower in low perturbation and larger in strong perturbation, see Fig 5 in DVERGE paper). This may also explain why higher transferability between sub-models is shown in Fig 3 as all of them learn robust features. Redoing Fig 3 with a smaller perturbation strength may unveil the difference in sub-model robustness between TRS and DVERGE, also a result with only the smoothness loss but not the cos loss would be helpful.

[1] Moosavi-Dezfooli, Seyed-Mohsen, et al. "Robustness via curvature regularization, and vice versa." Proceedings of the IEEE/CVF Conference on Computer Vision and Pattern Recognition. 2019.


In summary, I think this is a novel and solid theortical work supported by thorough experimental evaluation. Though there are concerns on whether the proposed smoothness objective is optimally designed and how the smoothness is contributing to the overall robustness exctly, I do think this can be a good starting point with high potential of inspiring future works on investigating how smoothness can help on achieving more robust ensemble models. So I would suggest accepting this submission, with the expectation that the authors can respond to my concerns in the discussion period.

**Time Spent Reviewing:**

2.5

---

> ### Author Response · Authors · 2021-08-10
> **Response to Reviewer 73Cj**
>
> We thank the reviewer for the insightful comments and provide our response as follows.
>
> >**Comment 1**: *One representative work is CURE [1], which also use a form of gradient regularization to minimize local curvature of deep learning model for robustness improvement. I think it would be important for the author to cite this paper and discuss the difference in the smoothness objective.*
>
> **Reply 1**: Thanks for your advice! We will cite this paper and make related discussions in our related work.
>
> >**Comment 2**: *Redoing Fig 3 with a smaller perturbation strength may unveil the difference in sub-model robustness between TRS and DVERGE, also a result with only the smoothness loss but not the cos loss would be helpful.*
>
> **Reply 2**: Thanks for your insightful comment. We redo the Fig 3 by decreasing the evaluation perturbation scale $\epsilon$ : $\epsilon = 0.05$ for MNIST (old: $0.3$), $\epsilon = 0.005$ for CIFAR-10 (old: $0.04$) following the suggestion. The comparison between TRS and DVERGE is shown in the following confusion matrices (the numbers in the cells represent the adversarial transfer rate, the lower the better):
>
> **The confusion matrix of the transferability on MNIST. Left: TRS; right: DVERGE**
>
> |      | TRS  |      |      |     |       | DVERGE |       |
> | :--: | :--: | :--: | :--: | :--: | :---: | :----: | :---: |
> |  44.8   |  **1.31**  | **1.79** |        |   | **15.7** |  2.56  | 2.04  |
> | 24.9   |  **3.50**   |   2.04   |      |    | **2.78**  | 18.4  |  **1.90**  |
> | 24.4   |  **1.50**   | **3.85** |      |    | **2.72**  |  2.36  | 13.8 |
>
> **The confusion matrix of the transferability on CIFAR-10. Left: TRS; right: DVERGE**
>
> |      | TRS  |      |      |     |       | DVERGE |       |
> | :--: | :--: | :--: | :--: | :--: | :---: | :----: | :---: |
> |  **44.8** |    14.9    |   17.0   |       |   | 53.9   |  **9.60**   | **10.2** |
> |16.2   |  **42.8**  |   15.3   |     |    | **11.6** |    53.6    | **10.2** |
> | 15.1   |    17.2    | **45.4** |     |    | **11.8** |  **9.60**   |   53.6   |
>
> We observe that some base models within the TRS ensemble indeed appear to be more robust compared to the DVERGE ensemble base models, which verified your hypothesis.
>
> We also did additional experiments by training with Lsmooth only. Results are shown as follows:
>
> | Robust acc on MNIST | &ensp;FGSM | &ensp;&ensp;BIM | &ensp;&ensp;PGD | &ensp;&ensp;MIM | &ensp;&ensp;CW | &ensp;&ensp;EAD |
> |---|:---:|:---:|:---:|:---:|:---:|:---:|
> | param. | $\epsilon=0.2$ | $\epsilon=0.15$ | $\epsilon=0.15$ | $\epsilon=0.15$ | $c=1.0$ | $c=10.0$ |
> | Lsim only | 30.7 | 0.0 | 0.0 | 0.0 | 58.6 | 0.5 |
> | Lsmooth only | **93.1** | 82.5 | 80.7 | 82.6 | 86.2 | 1.2 |
> | Lsim + Lsmooth (TRS) | 91.7 | **85.7** | **85.1** | **85.1** | **92.6** | **1.4** |
>
> | Robust acc on CIFAR-10 | &ensp;FGSM | &ensp;&ensp;BIM | &ensp;&ensp;PGD | &ensp;&ensp;MIM | &ensp;&ensp;CW | &ensp;&ensp;EAD |
> |---|:---:|:---:|:---:|:---:|:---:|:---:|
> | param. | $\epsilon=0.04$ | $\epsilon=0.02$ | $\epsilon=0.02$ | $\epsilon=0.02$ | $c=1.0$ | $c=5.0$ |
> | Lsim only              |          **35.0**          |          0.0          |          0.0          |          0.0          |     17.6     |       0.0       |
> | Lsmooth only           |          9.3           |         13.9          |         13.8          |         15.0          |     43.0     |       0.0       |
> | Lsim + Lsmooth (TRS)   |          24.9          |         **15.8**          |         **15.1**          |         **17.2**          |     **58.1**     |       **0.1**       |
>
> Results show that, though training with Lsmooth only could lead to high robustness, TRS ensemble could achieve even higher robustness against strong multi-step attacks by concerning similarity loss Lsim as well. This indicates that both model smoothness and model diversity are important.
>
> >**Comment 3**: *Provide detailed information on the hyperparameter (i.e. step size and the number of steps for the inner maximization) of the smoothness loss and discuss its impact on the overall training cost. If possible provide an ablation study on how these hyperparameters affect the final performance of TRS, is there a tradeoff in the training cost vs. robustness?*
>
> **Reply 3**: Thanks for the suggestion! We set step size $\alpha = \epsilon / 3$ for both MNIST and CIFAR datasets. The number of steps $T$ is set to be $6$ for MNIST and $5$ for CIFAR-10. We note that there is a linear correlation between training cost on computing Lsmooth and the number of steps $T$.
>
> Followed by the reviewer’s suggestions, for MNIST, we conducted additional experiments on surveying the tradeoff between training cost and robustness by varying the number of steps $T$ on solving the inner-maximization. Specifically, we considered the following settings:
>
> * $\alpha = \epsilon, T = 1$
> * $\alpha = \epsilon / 3, T = 6$ (default setting)
> * $\alpha = \epsilon / 10, T= 20$
>
> Results are shown as follows:
>
> | Robust acc on MNIST | &ensp;FGSM | &ensp;&ensp;BIM | &ensp;&ensp;PGD | &ensp;&ensp;MIM | &ensp;&ensp;CW | &ensp;&ensp;EAD |
> |---|:---:|:---:|:---:|:---:|:---:|:---:|
> | param. | $\epsilon=0.2$ | $\epsilon=0.15$ | $\epsilon=0.15$ | $\epsilon=0.15$ | $c=1.0$ | $c=10.0$ |
> | DVERGE                                |         91.6          |         47.7          |         28.8          |         44.6          |     79.2     |       0.0        |
> | TRS ($\alpha = \epsilon, T = 1$)      |         90.5          |         76.0          |         70.1          |         73.3          |     89.6     |       0.1        |
> | TRS ($\alpha = \epsilon / 3, T = 6$)  |         91.7          |         85.7          |         85.1          |         85.1          |     92.6     |     **1.4**      |
> | TRS ($\alpha = \epsilon / 10, T= 20$) |       **92.1**        |       **87.2**        |       **85.5**        |       **86.1**        |   **92.8**   |     **1.4**      |
>
> As we can see, TRS ensemble’s robustness consistently improves with the increasing of $T$ and converges. We should also notice that, even for $T = 1$, TRS ensemble is more robust than the strongest baseline DVERGE against various strong attacks. Due to the positive correlation between $T$ and the training cost, we should choose suitable $T$ balancing the training cost and model robustness. For MNIST, the default setting ($T = 6$) could be a good choice. We will add these results and discussion as a new ablation study subsection to our revision and thank you for the helpful suggestions.

---

> > ### Comment · Reviewer_73Cj · 2021-08-22
> > **Comments on the rebuttal**
> >
> > The rebuttal resolves my concern on the training cost, where the author show the optimization on the smoothness objective can converge in reasonable steps.
> >
> > As for the concern on smoothness objective contribution, I agree with the point raised by Reviewer zHq1, where the smoothness loss seems to be the main contributing factor on the overall ensemble robustness, rather than the diversity loss. Actually the smoothness loss may even hurt diversity to some extent given that it increase each sub-model's robustness. So there may be some overstatement in the contribution part. I would suggest the author to focus the contribution more on the theortical finding that smoothness loss is necessary to guarantee low attack transferability in diverse ensemble training (i.e. highlight the improvement of TRS over GAL), and clearly discuss how smoothness objective contribute to the ensemble robustness and affect the robustness of each individual sub-model.
> >
> > I would like to keep my score.

---

### Official Review · Reviewer_ABqv · 2021-07-16

**Rating:** 6
**Confidence:** 3

**Summary:**

This paper proposes to enforce the smoothness of the base models in addition to reducing the loss gradient similarity between them to improve ensemble model robustness. This is motivated by theoretical analysis for adversarial transferability and verified by extensive experiments.



**Limitations And Societal Impact:**

The author pointed out the limitation that "TRS ensemble is slightly less robust than DIVERGE under small perturbation with weak attack FGSM". If the method indeed significantly decreased the model performance over the normal dataset, it should also be noted as a limitation and potential negative societal impact.

**Main Review:**

To the best of my knowledge, the theoretical analysis and proposed method are novel. The experiments also support the central claim about improving ensemble model robustness.

However, the central concern I have is that: Is the proposed method essentially trading model performance for the "robustness"? After all, two completely different random guess models probably cannot transfer to each other, but such models are also useless. The proposed objective optimizes the base models to have smooth orthogonal decision boundaries, so that they are less adversarial examples are transferable between them. But this probably also makes at least one of them have much worse performance with such different decision boundaries as shown in figure 1. Then how can the ensemble still maintain the high performance?  On the other hand, previous works on promoting base model diversity to improve robustness still maintain that base models make the same predictions, i.e. having similar decision boundary, but potentially different confidence score distribution, such as in Improving Adversarial Robustness via Promoting Ensemble Diversity.

There are also several other issues
- It would be more convincing to test against stronger black-box attacks, such as
    - ILA (Enhancing adversarial example transferability with an intermediate level attack)
    - DI2-FGSM (Improving Transferability of Adversarial Examples with Input Diversity)
    - Interaction-Reduced attack (A Unified Approach to Interpreting and Boosting Adversarial Transferability)
- Standard deviations are not reported in table 1, but the error bar is claimed to be reported in the checklist.
- In Definition 4, the text on 159 suggesting that x and y should be paired input and true labels. But the notation in the formula could be interpreted otherwise. This should be clarified, maybe by qualifying (x,y) as a pair under the inf and sup. Similarly for other places applicable. In general, it would be better to use a different symbol for the qualified y that is potentially not the same as the ground truth label, such as the ones in Theorem 1.
-  On line 326, DIVERGE -> DVERGE




**Time Spent Reviewing:**

5

---

> ### Author Response · Authors · 2021-08-10
> **Response to Reviewer ABqv**
>
> We thank the reviewer for the insightful comments and provide our response as follows.
>
> >**Comment 1**: *... But this probably also makes at least one of them have much worse performance with such different decision boundaries as shown in figure 1. Then how can the ensemble still maintain the high performance? Is the proposed method essentially trading model performance for the "robustness"?*
>
> **Reply 1**:
> We thank the reviewer for the insightful question. We have reported the ensemble clean accuracy of TRS ensemble together with other baselines in Appendix G, Table 2 and 3. We can see that TRS ensemble only drops within 1% clean accuracy on MNIST, which is negligible. For CIFAR-10, TRS ensemble drops around 8% clean accuracy. This demonstrates that as an ensemble, TRS ensemble achieves both high (useful) benign accuracy, as well as the STOA robustness. Given the well-known tradeoff between robustness and benign performance [1], we believe the performance of TRS ensemble is promising and can inspire more robust algorithm design for the community.
>
> Concretely, we remark that the standard cross-entropy loss is still included in the TRS training to ensure the benign performance of base models. When some base models have low benign performance, the other base models can correct it and jointly bring high benign accuracy and robustness for the **whole TRS ensemble**.
>
> [1] Raghunathan, Aditi, et al. "Understanding and mitigating the tradeoff between robustness and accuracy." (2020).
>
> >**Comment 2**: *It would be more convincing to test against stronger black-box attacks, such as ILA, DI2-FGSM, Interaction-Reduced attack.*
>
> **Reply 2**: Thanks for your insightful suggestions. We evaluate these stronger black-box attacks on both MNIST and CIFAR-10 datasets. The reported **Robust accuracy (%)** under large perturbation scale $\epsilon$ are shown as follows:
>
> | MNIST ($\epsilon = 0.3$) |   ILA    | DI2-FGSM | Interaction-Reduced attack |
> | ------------------------ | :------: | :------: | :------------------------: |
> | ADP                      |   5.4    |   9.9    |            4.8             |
> | GAL                      |   3.0    |   8.6    |            7.1             |
> | DVERGE                   |   89.5   |   91.6   |            82.0            |
> | TRS                      | **91.2** | **93.7** |          **84.4**          |
>
> | CIFAR-10 ($\epsilon = 0.05$) |   ILA    | DI2-FGSM | Interaction-Reduced attack |
> | ---------------------------- | :------: | :------: | :------------------------: |
> | ADP                          |   1.2    |   1.6    |            1.4             |
> | GAL                          |   32.2   |   36.2   |            29.2            |
> | DVERGE                       |   35.9   |   38.3   |            32.4            |
> | TRS                          | **46.2** | **50.0** |          **45.1**          |
>
> All these black-box transfer attack images are generated from an unknown ensemble consisting of 3 ResNet20 models. It is clear that the TRS ensemble consistently demonstrates the highest robustness compared to other baselines. We will add these attack results and discussions in our revision.
>
> >**Comment 3**: *Standard deviations are not reported in table 1, but the error bar is claimed to be reported in the checklist.*
>
> **Reply 3**: Thanks for pointing this out. We re-evaluate our Table 1 and 2 by running attacks multiple times with different random seeds to obtain the standard deviation (within 1%) with consistent conclusions in the paper.  We will add these results in our revision.
>
> >**Comment 4**: *In Definition 4, the text on 159 suggesting that x and y should be paired input and true labels. But the notation in the formula could be interpreted otherwise. This should be clarified, maybe by qualifying (x,y) as a pair under the inf and sup. Similarly for other places applicable. In general, it would be better to use a different symbol for the qualified y that is potentially not the same as the ground truth label, such as the ones in Theorem 1.*
>
> **Reply 4**: Sorry for the confusion in the text. The equation is correct but the subsequent text description is imprecise - $y$ is not the true label but indeed could be any class label, i.e., the $x$ and $y$ are separately qualified. The reason is that since the attacker tries to alter the prediction class, the gradient diversity for different prediction classes should be considered. We will fix this typo in the revision, and we sincerely thank the reviewer for the nice suggestion!
>
> >**Comment 5**: *On line 326, DIVERGE -> DVERGE.*
>
> **Reply 5**: Thanks for the careful check! We will fix all the typos and make updates in our revision.
>
> We hope the reviewer can take our response into consideration and we are happy to provide additional evaluations and discussions if there are other suggestions!

---

### Official Review · Reviewer_gXTn · 2021-07-16

**Rating:** 6
**Confidence:** 3

**Summary:**

This work first theoretically analyzes and outlines sufficient conditions for adversarial transferability between models, then proposes a practical algorithm to reduce the transferability between base models within an ensemble to improve its robustness. They also provide the lower and upper bounds of adversarial transferability under certain conditions and propose an effective Transferability Reduced Smooth (TRS) ensemble training strategy to train a robust ensemble with low transferability. Extensive experiments demonstrate that the proposed TRS outperforms all baselines significantly.

**Limitations And Societal Impact:**

The theoretical analysis is provided but the practicability maybe weak.  I suggest to test the avaiablity in the real application.

**Main Review:**

[Strengths]
	+They make the first attempt towards a theoretical understanding of adversarial transferability, and provide an approach for developing robust ML ensembles.
	+A theoretical analysis is provided, which helps us to understand the transferability of adversarial examples between different models.
	+Extensive experiments demonstrate that the proposed TRS outperforms all baselines significantly. Ablation studies are provided.

[Weaknesses]
	-“white-box and blackbox attacks” is used in line 62, but “whitebox and blackbox attacks” is used in line 68, please write in the same way. “theoretical understanding” -> “a theoretical understanding”.
	-TRS requires smaller or comparable training time (line 65). But there is no statistical comparison of training times between different methods.
	-∥∇_x ˆ  l_F ∥_2 and ∥∇_x ˆ  l_G ∥_2 and used In Eq. (1). As I understand, it means for training a robust model B, model A is needed to calculate the smooth loss. In the end, model B may be “robust” for transferable adversarial examples from model A, but vulnerable for adversarial examples generated from other models. This type of training is limited in practical application.


**Time Spent Reviewing:**

five

---

> ### Author Response · Authors · 2021-08-10
> **Response to Reviewer gXTn**
>
> We thank the reviewer for the insightful comments and provide our response as follows.
>
> >**Comment 1**: *TRS requires smaller or comparable training time (line 65). But there is no statistical comparison of training times between different methods.*
>
> **Reply 1**: Thanks for pointing this out. We evaluated the average epoch training time for TRS and compared it to other baselines (ADP, GAL, DVERGE) on RTX 2080 single GPU device as follows:
>
> | Avg epoch training time (s) |  ADP  |  &ensp;GAL   | DVERGE |   &ensp; TRS   |
> | :-------------------------: |  :---: | :----: | :----: | :-----: |
> |            MNIST            |  29.22 | 106.81 | 184.42 | 302.24  |
> |          CIFAR-10           |  33.22 | 139.10 | 349.61 | 1291.55 |
>
> Our results show that though ADP, GAL require less training time, they can not achieve even comparable robustness with TRS as shown in our paper. Compared with DVERGE, TRS requires longer training time but maintains higher robustness under almost all attack scenarios.
>
>
> >**Comment 2**: *Model A is needed to calculate the smooth loss. In the end, model B may be “robust” for transferable adversarial examples from model A, but vulnerable for adversarial examples generated from other models. This type of training is limited in practical application.*
>
> **Reply 2**:
> We thank the reviewer for the insightful question, and we should emphasize that our goal is to develop robust ensemble models such that given an arbitrary adversarial image, at least a subset of base models could remain robust against it. That means, given an adversarial example generated based on other models, it either attacks some base models within the ensemble or cannot attack any of them. If it cannot attack any of the base models, the ensemble is trivially robust; If it attacks one or some of the base models, based on our transferability theoretical analysis, it cannot attack other base models within the ensemble with high probability, which means the whole ensemble will still be robust.
>
> Empirically, we observe that, TRS ensemble not only shows its high robustness against white-box attack, but also demonstrates its robustness against black-box transfer attack as shown in Figure 2 and Appendix G, where the adversarial images are generated from other unknown models.
>
> >**Comment 3**: *The theoretical analysis is provided but the practicability maybe weak. I suggest to test the availability in the real application.*
>
> **Reply 3**: Thanks for your suggestion. Empirically, we follow the standard robustness evaluation (white-box, black-box, transferability) on real-world datasets (MNIST, CIFAR-10, CIFAR-100), and compare TRS ensemble with several stoa robust ensemble baselines based on their settings. We will add corresponding discussions for our empirical evaluation in the revision.

---

### Official Review · Reviewer_zHq1 · 2021-07-17

**Rating:** 6
**Confidence:** 4

**Summary:**

The manuscript describes the problem of adversarial transferability, the process of adversarial attacks that can be transferred across models, which can help scale adversarial attack strategies. As a guardrail against such attacks, it is desired that models are robust and protected against such attacks. Towards solving this, the authors first derive a lower and upper bound on a transferability metric that they define. The similarity between gradient losses was found to be the salient cross-model term affecting adversarial transferability. Subsequently, this pointed to a strategy of reducing model smoothness and loss gradient similarity across models as a way to reduce adversarial transferability.

**Limitations And Societal Impact:**

The authors do not discuss limitations of their approach. Is truncating the Taylor series with the first term correct? What about higher order terms, which might come into play when source models are unknown. The methods developed in general will have a positive societal impact by reducing adversarial attacks.

**Main Review:**

Motivated by reducing model smoothness, the authors propose a Transferability Reduced Smoothness (TRS) approach. The authors define adversarial attacks, both untargeted (goal is to mislabel) and targeted (goal is to specify an incorrect label). The definitions 1-5 and derivations are clear. The two central metrics driving the developments in the paper are the loss gradient similarity, which describes similarity between the gradient of losses for two models, and a beta-smoothness, which defines when a model is beta-smooth.

The two loss functions proposed, for smoothness and similarities, are fairly novel and appear to work well for the benchmarks. My two big concerns are that the experiments do not have the same level of clarity and detail as the theoretical counterpart, and the real-world use case of preventing adversarial attacks from an unknown source model are not addressed. My detailed comments below:

One key weakness in the utility of such a framework is that the training is across pairs of models while in reality the adversarial transferability has to be against unknown source models. I believe the authors should also look at source model agnostic solutions to make their proposed work useful in real-world settings.

“Luckily, inspired from deep learning theory and optimization [14, 36], succinct l2 regularization on the gradient terms can reduce the magnitude of gradients and thus improve model smoothness.“ This statement is not well supported and appears to be a very crude approximation to the beta-smoothness metric (Definition 4). Equation 1 only has weak similarity between the two models F and G.
Which of Lsim or Lsmooth is contributing more to the losses?

The real meat of the paper starts only on page 5. I think the authors should condense pages 1-4 to within three pages to get to the new developments of the paper faster.

Experimental results section is rushed and comprises only 1 page. It is concerning that in Fig. 3, the results of the proposed method for some model pairs are significantly lower than the DVERGE baseline method.

I have read the authors' rebuttals as well as the other reviews.


**Time Spent Reviewing:**

2

---

> ### Author Response · Authors · 2021-08-10
> **Response to Reviewer zHq1**
>
> We thank the reviewer for the insightful comments and provide our response as follows.
>
> >**Comment 1**: *One key weakness in the utility of such a framework is that the training is across pairs of models while in reality the adversarial transferability has to be against unknown source models. I believe the authors should also look at source model agnostic solutions to make their proposed work useful in real-world settings.*
>
> **Reply 1**: We thank the reviewer for the comment, and we should emphasize that our proposed robust ensemble is indeed source model agnostic both theoretically and empirically.
> Concretely, given an adversarial example generated based on other models, it either attacks some base models within the ensemble or cannot attack any of them. If it cannot attack any of the base models, the ensemble is trivially robust; If it attacks one or some of the base models, based on our transferability theoretical analysis, it cannot attack other base models within the ensemble with high probability, which means the whole ensemble will still be robust.
>
> Empirically, we observe that, TRS ensemble not only shows its high robustness against white-box attack, but also demonstrates its robustness against black-box transfer attack as shown in Figure 2 and Appendix G, where the adversarial images are generated from other unknown source models.
>
> As a summary, we theoretically and empirically show that constraining adversarial transferability between base models **within** the ensemble is sufficient as a defense for ensemble, which is also the methodology of related works [1,2] only empirically. We believe our proposed method is practical in real-world settings given its efficient and effective robustness performance.
>
> >**Comment 2**: *“Succinct l2 regularization on the gradient terms can reduce the magnitude of gradients and thus improve model smoothness.“ This statement is not well supported and appears to be a very crude approximation to the beta-smoothness metric (Definition 4).*
>
> **Reply 2**:
> We thank the reviewer for pointing this out. Corollary 4 of [3] shows an explicit connection between the (maximum) magnitude of gradients (which is equivalent to first-order Lipschitz bound $\alpha_L(\theta)$) and the smoothness $\beta$: $\beta = O(\alpha_L(\theta)^2)$. We will cite this paper and add discussion in our revision to make it clear and motivate the $\ell_2$ regularization.
>
> >**Comment 3**: *Which of Lsim or Lsmooth is contributing more to the losses?*
>
> **Reply 3**: Thanks for the insightful question. From our results (Table 1 and 2, Fig 3), we show that training with only similarity loss can not provide enough robustness against strong attacks. We follow the suggestion and additionally evaluate Lsim only and Lsmooth only trained ensembles against various white-box attacks as below:
>
> | Robust acc on MNIST | &ensp;FGSM | &ensp;&ensp;BIM | &ensp;&ensp;PGD | &ensp;&ensp;MIM | &ensp;&ensp;CW | &ensp;&ensp;EAD |
> |---|:---:|:---:|:---:|:---:|:---:|:---:|
> | param. | $\epsilon=0.2$ | $\epsilon=0.15$ | $\epsilon=0.15$ | $\epsilon=0.15$ | $c=1.0$ | $c=10.0$ |
> | Lsim only | 30.7 | 0.0 | 0.0 | 0.0 | 58.6 | 0.5 |
> | Lsmooth only | **93.1** | 82.5 | 80.7 | 82.6 | 86.2 | 1.2 |
> | Lsim + Lsmooth (TRS) | 91.7 | **85.7** | **85.1** | **85.1** | **92.6** | **1.4** |
>
> | Robust acc on CIFAR-10 | &ensp;FGSM | &ensp;&ensp;BIM | &ensp;&ensp;PGD | &ensp;&ensp;MIM | &ensp;&ensp;CW | &ensp;&ensp;EAD |
> |---|:---:|:---:|:---:|:---:|:---:|:---:|
> | param. | $\epsilon=0.04$ | $\epsilon=0.02$ | $\epsilon=0.02$ | $\epsilon=0.02$ | $c=1.0$ | $c=5.0$ |
> | Lsim only              |          **35.0**          |          0.0          |          0.0          |          0.0          |     17.6     |       0.0       |
> | Lsmooth only           |          9.3           |         13.9          |         13.8          |         15.0          |     43.0     |       0.0       |
> | Lsim + Lsmooth (TRS)   |          24.9          |         **15.8**          |         **15.1**          |         **17.2**          |     **58.1**     |       **0.1**       |
>
> Our results shows that: Both Lsim and Lsmooth is contributing to the ensemble robustness. Normally, Lsmooth-only trained model achieves higher robustness than Lsim-only, i.e., Lsmooth may contribute more. By combining Lsim and Lsmooth for training, we can achieve the highest robustness.
>
> >**Comment 4**: *I think the authors should condense pages 1-4 to within three pages to get to the new developments of the paper faster.*
>
> **Reply 4**: Thanks for the suggestion! We will definitely condense pages 1-4 and illustrate the method and experiments with more details.
>
> >**Comment 5**: *It is concerning that in Fig. 3, the results of the proposed method for some model pairs are significantly lower than the DVERGE baseline method.*
>
> **Reply 5**: Thanks for pointing this out. We believe that the TRS method still reduces the transferability significantly, especially when compared with ADP and GAL. The relatively smaller improvement on transferability compared with DVERGE may be a side-effect of smoothness regularization. However, with both Lsim and Lsmooth, overall, TRS provides a significantly higher robustness under white-box and black-box attacks. Since the real-world robustness is more important than transferability itself which is a proxy of the final robustness of the ensemble, TRS clearly provides a more robust ensemble compared with stoa baselines.
>
> >**Comment 6**: *Is truncating the Taylor series with the first term correct? What about higher order terms, which might come into play when source models are unknown.*
>
> **Reply 6**: When the smoothness of the two models are bounded by $\beta$, we can bound the higher-order terms in Taylor series by Lagrangian error bound [4]: $|err| \le \frac{1}{2} \beta \epsilon^2$ where $\epsilon$ is the perturbation radius.
> For unknown models, since the smoothness cannot be bounded, the higher-order terms cannot be bounded. However, as we illustrated, only considering adversarial transferability between base models **within** the ensemble is sufficient given the finite possible transferability directions shown in existing work [5].
> We will discuss more societal impacts of this work in revision, and since we are proposing a “robust” ensemble against potential adversarial attacks, our work has no negative social impact.
>
> [1] Pang, Tianyu, et al. "Improving adversarial robustness via promoting ensemble diversity." ICML 2019.
>
> [2] Yang, Huanrui, et al. DVERGE: diversifying vulnerabilities for enhanced robust generation of ensembles. NeurIPS 2020.
>
> [3] Sinha, Aman, Hongseok Namkoong, and John Duchi. "Certifying Some Distributional Robustness with Principled Adversarial Training." International Conference on Learning Representations. 2018.
>
> [4] https://www.expii.com/t/lagrange-error-bound-321
>
> [5] Tramèr, Florian, et al. "The space of transferable adversarial examples." (2017).

---

### Decision · Program_Chairs · 2021-09-27

**Decision:**

Accept (Poster)

**Comment:**

This paper analyzes the reason for adversarial transferability between models, and proposes an algorithm to reduce the transferability between base models. All reviewer agree that this is an interesting topic and the paper provides theoretical insight on the transferability problem. On the other side, the experimental analyses need to be strengthened. What are the additional costs for the reduction of adversarial transferability? Overall, this is an interesting paper. I recommend accept.